# DISTILLING COGNITIVE BACKDOOR PATTERNS WITHIN AN IMAGE

**Hanxun Huang[1]   Xingjun Ma[2]† Sarah Erfani[1]   James Bailey[1]**

[1]School of Computing and Information Systems, The University of Melbourne, VIC, Australia
[2]School of Computer Science, Fudan University, Shanghai, China
`hanxunh@student.unimelb.edu.au; xingjunma@fudan.edu.cn;`
`sarah.erfani@unimelb.edu.au; baileyj@unimelb.edu.au.`

## ABSTRACT

This paper proposes a simple method to distill and detect backdoor patterns within an image: *Cognitive Distillation* (CD). The idea is to extract the "minimal essence" from an input image responsible for the model's prediction. CD optimizes an input mask to extract a small pattern from the input image that can lead to the same model output (i.e., logits or deep features). The extracted pattern can help understand the cognitive mechanism of a model on clean vs. backdoor images and is thus called a *Cognitive Pattern* (CP). Using CD and the distilled CPs, we uncover an interesting phenomenon of backdoor attacks: despite the various forms and sizes of trigger patterns used by different attacks, the CPs of backdoor samples are all surprisingly and suspiciously small. One thus can leverage the learned mask to detect and remove backdoor examples from poisoned training datasets. We conduct extensive experiments to show that CD can robustly detect a wide range of advanced backdoor attacks. We also show that CD can potentially be applied to help detect potential biases from face datasets. Code is available at `https://github.com/HanxunH/CognitiveDistillation`.

## 1 INTRODUCTION

Deep neural networks (DNNs) have achieved great success in a wide range of applications, such as computer vision (He et al., 2016a; Dosovitskiy et al., 2021) and natural language processing (Devlin et al., 2019; Brown et al., 2020). However, recent studies have shown that DNNs are vulnerable to backdoor attacks, raising security concerns on their deployment in safety-critical applications, such as facial recognition (Sharif et al., 2016), traffic sign recognition (Gu et al., 2017), medical analysis (Feng et al., 2022), object tracking (Li et al., 2022), and video surveillance (Sun et al., 2022). A backdoor attack implants a backdoor trigger into the target model by poisoning a small number of training samples, then uses the trigger pattern to manipulate the model's predictions at test time. A backdoored model performs normally on clean test samples, yet consistently predicts the backdoor label whenever the trigger pattern appears. Backdoor attacks could happen in scenarios where the datasets or pre-trained weights downloaded from unreliable sources are used for model training, or training a model on a Machine Learning as a Service (MLaaS) platform hosted by an untrusted party.

Backdoor attacks are highly stealthy, as 1) they only need to poison a few training samples; 2) they do not affect the clean performance of the attacked model; and 3) the trigger patterns are increasingly designed to be small, sparse, sample-specific or even invisible. This makes backdoor attacks hard to detect or defend without knowing the common characteristics of the trigger patterns used by different attacks, or understanding the cognitive mechanism of the backdoored models in the presence of a trigger pattern. To address this challenge, trigger recovery and mitigation methods, such as Neural Cleanse (NC) (Wang et al., 2019), SentiNet (Chou et al., 2020) and Fine-Pruning (Liu et al., 2018a), have been proposed to reverse engineer and remove potential trigger patterns from a backdoored model. While these methods have demonstrated promising results in detecting backdoored models, they can still be evaded by more advanced attacks (Nguyen & Tran, 2020; Li et al., 2021c).

---

† Corresponding author.

Several works have aimed to shed light on the underlying backdoor vulnerability of DNNs. It has been shown that overparameterized DNNs have the ability to memorize strong but task-irrelevant correlations between a frequently appearing trigger pattern and a backdoor label (Gu et al., 2017; Geirhos et al., 2020). In fact, training on a backdoor-poisoned dataset can be viewed as a dual-task learning problem, where the clean samples define the *clean task* and the backdoor samples define the *backdoor task* (Li et al., 2021a). DNNs can learn both tasks effectively and in parallel without interfering (too much) with each other. However, the two tasks might not be learned at the same pace, as it has been shown that DNNs converge much faster on backdoor samples (Bagdasaryan & Shmatikov, 2021; Li et al., 2021a).

Other studies suggest that backdoor features are outliers in the deep representation space (Chen et al., 2018; Tran et al., 2018). Backdoor samples are also input-invariant, i.e., the model's prediction on a backdoor sample does not change when the sample is mixed with different clean samples (Gao et al., 2019). It needs a smaller distance to misclassify all samples into the backdoor class (Wang et al., 2019), and backdoor neurons (neurons that are more responsive to the trigger pattern) are more sensitive to adversarial perturbations (Wu & Wang, 2021). While the above findings have helped the development of a variety of backdoor defense techniques, the cognitive mechanism of how the predictions of the attacked model are hijacked by the trigger pattern is still not clear.

In this paper, we propose an input information disentangling method called *Cognitive Distillation* (CD) to distill a minimal pattern of an input image determining the model's output (e.g. features, logits and probabilities). The idea is inspired by the existence of both useful and non-useful features within an input image (Ilyas et al., 2019). Intuitively, if the non-useful features are removed via some optimization process, the useful features will be revealed and can help understand the hidden recognition mechanism for the original input. CD achieves this by optimizing an input mask to remove redundant information from the input, whilst ensuring the model still produces the same output. The extracted pattern is called a *Cognitive Pattern* (CP) and intuitively, it contains the *minimum sufficient* information for the model's prediction.

Using CD, we uncover an interesting phenomenon of backdoor attacks: the CPs of backdoor samples are all surprisingly and suspiciously smaller than those of clean samples, despite the trigger patterns used by most attacks spanning over the entire image. This indicates that the backdoor correlations between the trigger patterns and the backdoor labels are much simpler than the natural correlations. So small trigger patterns may be sufficient for effective backdoor attacks. This common characteristic of existing backdoor attacks motivates us to leverage the learned masks to detect backdoor samples. Moreover, the distilled CPs and learned masks visualize how the attention of the backdoored models is shifted by different attacks.

Our main contributions are summarized as follows:

- We propose a novel method *Cognitive Distillation* (CD) to distill a minimal pattern within an input image determining the model's output. CD is self-supervised and can potentially be applied to any type of DNN to help understand a model's predictions.
- Using CD, we uncover a common characteristic of backdoor attacks: the CPs of backdoor samples are generally smaller than those of clean samples. This suggests that backdoor features are simple in nature and the attention of a backdoored model can be attracted by a small part of the trigger patterns.
- We show that the $L_1$ norm of the learned input masks can be directly used to not only detect a wide range of advanced backdoor attacks with high AUROC (area under the ROC curve), but also help identify potential biases from face datasets.

## 2 RELATED WORK

We briefly review related works in backdoor attack and defense. Additional technical comparison of our CD with related methods can be found in Appendix A.

**Backdoor Attack.** The goal of backdoor attacks is to trick a target model to memorize the backdoor correlation between a trigger pattern and a backdoor label. It is closely linked to the overparameterization and memorization properties of DNNs. Backdoor attacks can be applied under different threat models with different types of trigger patterns. Based on the adversary's knowledge, existing

backdoor attacks can be categorized into *data-poisoning attacks* and *training-manipulation attacks*. Data-poisoning attacks inject the trigger pattern into a few training samples but do not have access to the training process, whereas training-manipulation attacks can access and modify the training data, procedure, and objective function to implant the trigger (Garg et al., 2020; Lin et al., 2020; Shumailov et al., 2021; Bagdasaryan & Shmatikov, 2021; Nguyen & Tran, 2021; Doan et al., 2021a;b).

The focus of our work is data-poisoning attacks. Early works, such as BadNets (Gu et al., 2017), Blend (Chen et al., 2017) and Trojan (Liu et al., 2018b), use simple trigger patterns like black-white square, patch trigger or blending background to backdoor DNNs. Later works propose more complex trigger patterns, such as periodical signal pattern (SIG) (Barni et al., 2019), simulated natural reflections (Refool) (Liu et al., 2020), generative adversarial networks (GAN) generated patterns (for time series data) (Jiang et al., 2023), or physical world patterns/objects (Li et al., 2020; Wenger et al., 2021), to achieve more powerful attacks. One can also utilize adversarial perturbations (Turner et al., 2018; Zhao et al., 2020b), Instagram filters (Liu et al., 2019), smooth image frequency (Zeng et al., 2021) or GANs (Cheng et al., 2020) to boost the strength of the trigger patterns. Besides the dataset-wise or class-wise triggers used by the above attacks, more recent works leverage sample-specific (Nguyen & Tran, 2020) and invisible (Li et al., 2021c) trigger patterns to craft more stealthy attacks. In this work, we will show one common characteristic of the above attacks that is related to their trigger patterns.

**Backdoor Defense.** Existing backdoor defense methods can be categorized into: 1) trigger recovery, 2) backdoor model detection, 3) backdoor sample detection, and 4) mitigation methods, where the first three types of methods are oftentimes required for mitigation. Trigger recovery aims to reverse engineer the trigger pattern (Wang et al., 2019; Guo et al., 2019; Liu et al., 2019; Sun et al., 2020; Liu et al., 2022; Xiang et al., 2022; Hu et al., 2022). Backdoor model detection aims to determine if a model is affected by triggers (Chen et al., 2019; Kolouri et al., 2020; Wang et al., 2020; Guo et al., 2021; Shen et al., 2021; Xu et al., 2021), It is worth noting that the detected models still need mitigation methods to remove the trigger (Liu et al., 2018a; Zhao et al., 2020a; Wu & Wang, 2021; Li et al., 2021b; Zeng et al., 2022; Guan et al., 2022). Backdoor mitigation can also be achieved by robust learning strategies (Borgnia et al., 2021; Huang et al., 2022; Dolatabadi et al., 2022).

Backdoor sample detection assesses if a sample is a backdoor sample, i.e., whether it contains a trigger pattern. The backdoor images may show anomalies in the frequency domain but could be hidden by the attacker using smoothing (Zeng et al., 2021). Spectral Signatures (SS) uses deep feature statistics to discriminate between clean and backdoor samples (Tran et al., 2018), but it is less robust to the change of poisoning rate (Hayase et al., 2021). Feature-based detection can also be performed via identity-variation decomposition (Tang et al., 2021), activation clustering (AC) (Chen et al., 2018), and feature consistency towards transformations (FCT) (Chen et al., 2022). STRIP proposes a superimposition technique to blend the potentially backdoored samples with a small subset of clean samples, then utilizes the entropy of the predictions for detection (Gao et al., 2019). Anti-Backdoor Learning (ABL) monitors sample-specific training loss to isolate low-loss backdoor samples (Li et al., 2021a). It has also been theoretically proven that, under a restricted poisoning rate, robust learning on poisoned data is equivalent to the detection and removal of corrupted points (Manoj & Blum, 2021). This highlights the importance of backdoor sample detection where our proposed CD can be applied.

## 3 COGNITIVE DISTILLATION AND BACKDOOR SAMPLE DETECTION

We first introduce our CD method, then present our findings on the backdoored models and the proposed backdoor sample detection method.

### 3.1 COGNITIVE DISTILLATION

Given a DNN model $f_\theta$ and an input image $\boldsymbol{x} \in \mathcal{X} \subset \mathbb{R}^{w \times h \times c}$ ($w$, $h$, $c$ are the width, height and channel, respectively), CD learns an input mask to distill a minimal pattern from $\boldsymbol{x}$ by solving the following optimization problem:

$$\arg\min_{\boldsymbol{m}} \|f_\theta(\boldsymbol{x}) - f_\theta(\boldsymbol{x}_{cp})\|_1 + \alpha\|\boldsymbol{m}\|_1 + \beta TV(\boldsymbol{m}) \tag{1}$$

$$\boldsymbol{x}_{cp} = \boldsymbol{x} \odot \boldsymbol{m} + (1 - \boldsymbol{m}) \odot \delta, \tag{2}$$

where, $x_{cp} \in \mathcal{X} \subset \mathbb{R}^{w \times h \times c}$ is the distilled cognitive pattern, $m \in [0, 1]^{w \times h}$ is a learnable 2D input mask that does not include the color channels, $\delta \in [0, 1]^c$ is a $c$-dimensional random noise vector, $\odot$ is the element-wise multiplication applied to all the channels, $TV(\cdot)$ is the total variation loss, $\|\cdot\|_1$ is the $L_1$ norm, $\alpha$ and $\beta$ are two hyperparameters balancing the three terms. The model output $f_\theta(\cdot)$ can be either logits (output of the last layer) or deep features (output at the last convolutional layer).

The first term in Equation 1 ensures the model's outputs are the same on the distilled pattern $x_{cp}$ and the original input $x$; the second term enables to find small (sparse) cognitive patterns and remove the non-useful features from the input; the third TV term regularizes the mask to be smooth. In Equation 2, a mask value $m_i$ close to 1 means the pixel is important for the model's output and should be kept, close to 0, otherwise. Here, instead of directly removing the unimportant pixels, we use uniformly distributed random noise (i.e, $\delta$) to replace them *at each optimization step*. This helps to distinguish important pixels that originally have 0 values.

By optimizing Equation 1, we can obtain a 2D mask $m$ and a CP $x_{cp}$, where the mask highlights the locations of the important pixels, and the CP is the extracted pattern. It is worth noting that a CP is a perturbed pattern from the input image, which may not be the raw (original) pixels. The masks and CPs extracted for different models can help to understand their prediction behaviors.

## 3.2 UNDERSTANDING BACKDOORED MODELS WITH CD

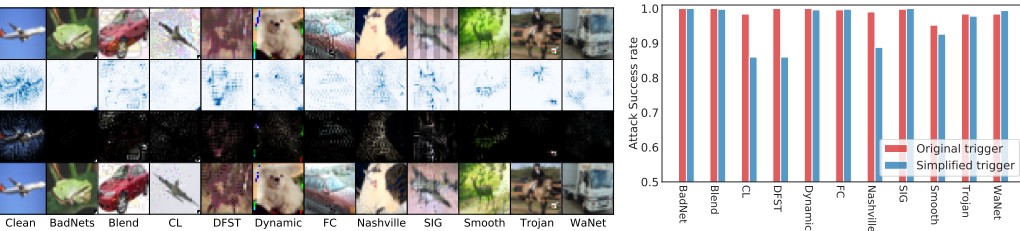

(a) Mask and CP visualizations                    (b) Attack with simplified triggers

Figure 1: (a) **First row**: a clean image and example backdoored images. **Second row**: the corresponding learned masks. **Third row**: distilled cognitive patterns. **Fourth row**: simplified backdoor images. The studied backdoor attacks include BadNets (Gu et al., 2017), Blend (Chen et al., 2017), CL (Turner et al., 2018), DFST (Cheng et al., 2020), Dynamic (Nguyen & Tran, 2020), FC (Shafahi et al., 2018), Nashville (Liu et al., 2019), SIG (Barni et al., 2019), Smooth (Zeng et al., 2021), Trojan (Liu et al., 2018b) and WaNet (Nguyen & Tran, 2021). (b) The attack success rate of the simplified (the Fourth row in (a)) and the original triggers.

We first apply CD to understand the inference mechanism of backdoored models on clean vs. backdoor samples. The experiments are conducted with ResNet-18 (He et al., 2016a) on CIFAR-10 (Krizhevsky et al., 2009) dataset. We apply backdoor attacks (listed in Figure 1) with a poisoning rate of 5% to obtain the backdoored models. We then distill the clean CP of a clean training image on a backdoored model (BadNets), and the backdoor CPs are distilled on the backdoored training images with corresponding models. Examples of the distilled CPs and their corresponding masks are visualized in Figure 1a. More visualizations for all the attacks can be found in Appendix D.

As shown in the first column of Figure 1a, the mask and CP of the clean image on a backdoored model are large and semantically associated with the main object. Clearly, the model appears to be using the real content within the image to make the prediction. For the 3 attacks (BadNets, Trojan, and Dynamic) that use small patches or scattered pixels as trigger patterns, the CPs of the backdoor images reveal their trigger patterns and the masks highlight the key part of the triggers.

The other 8 attacks, all adopt full-image size triggers, yet their CPs are all suspiciously small (in magnitude), sparse (scattered pixels), and semantically meaningless (drifting away from the main object). This suggests that the model is indeed using the backdoor features to predict the class label. Interestingly, it shows that only a small part of the trigger pattern is involved in the inference process, even if the trigger spans over the entire image. It seems that the model not only ignores the real content but also a large part of the trigger. We conjecture that this is because backdoor correlations are simpler in nature when compared with natural correlations, thus the model does not need to memorize the entire trigger pattern to learn the backdoor correlation. Rather, it tends to find

the simplest clues from the trigger pattern to complete the backdoor task. These simple correlations bypass the perception of the real content at inference time whenever the trigger appears.

Next, we run a set of experiments to confirm that simpler triggers extracted by our CD can work the same as the original triggers of the above attacks. A backdoor training sample with a simplified trigger can be generated by:

$$\boldsymbol{x}'_{bd} = \boldsymbol{m} \odot \boldsymbol{x}_{bd} + (1 - \boldsymbol{m}) \odot \boldsymbol{x}, \tag{3}$$

where, $\boldsymbol{x}_{bd}$ is the original backdoor sample, $\boldsymbol{x}'_{bd}$ is the backdoor sample with simplified trigger, $\boldsymbol{m}$ is a binarized (with threshold 0.05) version of the learned mask by our CD, and $\boldsymbol{x}$ is the clean sample. For the 8 full-image size attacks, the trigger size is reduced from 100% to $\sim 40\%$ of the image size. Note that the reduced part is replaced by the original clean pixels and the test triggers remain unchanged. The attack success rate (ASR) of the simplified triggers is reported and compared with the original triggers in Figure 1b. It is evident that every trigger can be simplified without losing (much) of the ASR. Interestingly, the simplified trigger even slightly improves the ASRs of FC, SIG, and WaNet attacks.

To summarize, the above findings reveal one common characteristic of backdoor attacks: backdoor correlations are much simpler than natural correlations, regardless of the trigger patterns. One can thus utilize the size of the distilled masks by our CD to detect backdoor samples.

## 3.3 BACKDOOR SAMPLE DETECTION

Based on the assumption that the model predictions on backdoored images will rely on a small set of pixels, we introduce CD as a backdoor sample detection method. Detection can be performed at either training or test time. At training time, the defender can remove backdoor samples from the training dataset, while at test time, the defender can expose potential attacks and their triggers.

**Threat Model.** Following previous works (Tran et al., 2018; Gao et al., 2019; Li et al., 2021a), we assume the adversary can poison the defender's training data but does not have access to the training process. We also assume the defender has full control over the training process but has no prior knowledge of i) the poisoning rate, ii) the trigger pattern, iii) the backdoored class, or iv) whether a sample is clean or backdoored.

**Problem Formulation.** Considering a $K$-class image classification task, we denote the training data as $\mathcal{D} = \mathcal{D}_c \cup \mathcal{D}_b$, the clean subset as $\{(\boldsymbol{x}_i, y_i)\}_{i=1}^N \in \mathcal{D}_c$, and the poisoned subset as $\{(\boldsymbol{x}_i^{bd}, y_i^{bd})\}_{i=1}^M \in \mathcal{D}_b$, respectively. The attacker injects backdoor triggers using function $(\boldsymbol{x}_i^{bd}, y_i^{bd}) = A(\boldsymbol{x}_i, y_i)$, which converts a clean sample into a backdoor sample. The $\boldsymbol{x} \in \mathcal{X} \subset \mathbb{R}^{w \times h \times c}$ are the inputs and $y \in \mathcal{Y} = \{1, \cdots, K\}$ are the labels for $K$ classes in total. The poisoning rate is defined as $\frac{|\mathcal{D}_b|}{|\mathcal{D}|} = \frac{M}{M+N}$. The defender's goal is to accurately detect samples $\boldsymbol{x} \in \mathcal{D}_b$.

Backdoor sample detection is an unsupervised binary classification task (backdoor class and clean class). We denote the predicted backdoor samples as $\mathcal{D}'_b$ and the predicted clean samples as $\mathcal{D}'_c$. Based on our finding that the CPs of backdoor samples are suspiciously small, here we propose to use the $L_1$ norm of the learned mask $\boldsymbol{m}$ to detect backdoor samples, as it measures the pixel intensity of the CP distilled by CD. We consider the following function $g_i$ to determine whether a sample $\boldsymbol{x}$ contains backdoor based on its mask $\boldsymbol{m}$:

$$g(\boldsymbol{x}) = \begin{cases} 1 & \text{if } \|\boldsymbol{m}\|_1 \le t, \\ 0 & \text{if } \|\boldsymbol{m}\|_1 > t, \end{cases} \tag{4}$$

where, $t$ is a threshold, $g(\cdot) = 1$ indicates a backdoor sample, whereas $g(\cdot) = 0$ indicates a clean sample. While other strategies (e.g., training a detector on the masks or distilled CPs) are also plausible, the above simple thresholding strategy proves rather effective against a wide range of backdoor attacks. The distribution of $\|\boldsymbol{m}\|_1$ in Figure 2 confirms the separability of backdoor samples from the clean ones. In practice, the threshold $t$ can be flexibly determined based on the distribution of $\|\boldsymbol{m}\|_1$, for example, detecting samples of significantly lower $\|\boldsymbol{m}\|_1$ than the mean or median as backdoor samples.

For test time detection, we assume the defender can access a small subset of confirmed clean samples $\mathcal{D}_s$. The defender can then calculate the distribution, the mean $\mu_{\|\boldsymbol{m}\|_1}$, and the standard deviation $\sigma_{\|\boldsymbol{m}\|_1}$ for $\boldsymbol{x}_i \in \mathcal{D}_s$. The threshold can then be determined as: $t = \mu_{\|\boldsymbol{m}\|_1} - \gamma \cdot \sigma_{\|\boldsymbol{m}\|_1}$, where $\gamma$ is a hyperparameter that controls the sensitivity.

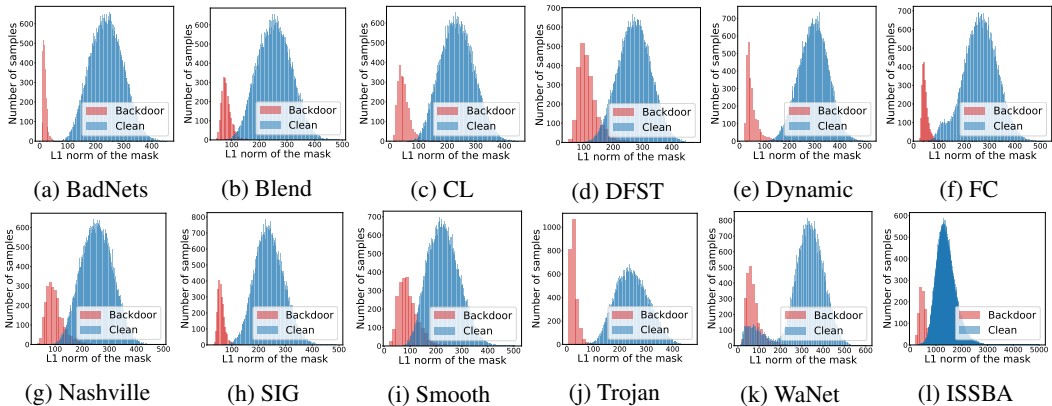

Figure 2: The distribution of $\|\boldsymbol{m}\|_1$ for clean (blue) and backdoor (red) samples crafted by different backdoor attacks. **X-axis**: $L_1$ norm of the learned mask ($\|\boldsymbol{m}\|_1$); **Y-axis**: the number of samples.

## 4 EXPERIMENTS

In this section, we evaluate our CD-based detection method in terms of detection performance, robustness under different poisoning rates, and the improvement it brings to backdoor mitigation. We first describe the experimental setup and then present the results in Section 4.1 - 4.3. A detailed ablation study on the hyperparameters and optimization terms of CD can be found in Appendix B.5 and B.6. It shows that CD-based detection is insensitive to the $\beta$ hyperparameter and is moderately stable under varying $\alpha$. We also provide an analysis of the detection performance under three types of adaptive attacks in Appendix B.7. We find that the attacker needs to sacrifice two crucial elements (attack success rate or stealthiness) to partially evade our detection method.

**Attack Configurations.** We consider 12 advanced backdoor attacks, including: BadNets (Gu et al., 2017), Blend (Chen et al., 2017), CL (Turner et al., 2018), DFST (Cheng et al., 2020), Dynamic (Nguyen & Tran, 2020), FC (Shafahi et al., 2018), SIG (Barni et al., 2019), Trojan (Liu et al., 2018b),, using WaNet trigger (Nguyen & Tran, 2021), Nashivell filter (Liu et al., 2019) Smooth (Zeng et al., 2021) and ISSBA (Li et al., 2021c). We perform evaluations on 3 datasets, including CIFAR-10 (Krizhevsky et al., 2009), an ImageNet (Deng et al., 2009) subset (200 classes), and GTSRB (Houben et al., 2013). No data augmentations are applied during training as it may decrease the attack performance (Liu et al., 2020). We adopt 6 types of architectures, including VGG-16 (Simonyan & Zisserman, 2015), ResNet-18 (RN-18)(He et al., 2016a), PreActivationResNet-101 (PARN-101) (He et al., 2016b), MobileNetV2 (MobileV2) (Sandler et al., 2018), GoogLeNet (Szegedy et al., 2015), and EfficientNet-b0 (Tan & Le, 2019). More details are in Appendix B.2.

**Detection Configurations.** We compare our CD approach with 5 state-of-the-art backdoor sample detection methods: AC (Chen et al., 2018), SS (Tran et al., 2018), STRIP (Gao et al., 2019), Frequency Zeng et al. (2021), and the isolation strategy used in ABL (Li et al., 2021a). For test time detection, we use the model's prediction as the label for AC and SS and exclude ABL from this experiment as it cannot be applied at test time. For our CD method, we use both the logits layer and the deep feature layer (the last activation layer) as the model output and denote the respective method as **CD-L** and **CD-F**. More details are in Appendix B.3.

**Evaluation Metrics.** We adopt the area under the area under the ROC curve (AUROC) as the main evaluation metric. Following (Gao et al., 2019), we also consider the true rejection rate (TRR) and false acceptance rate (FAR) as supplementary performance metrics. TRR is defined as the proportion of backdoor samples in $\mathcal{D}'_b$, whilst FAR is defined as the proportion of backdoor samples in $\mathcal{D}'_c$.

### 4.1 DETECTION PERFORMANCE EVALUATION

Table 1 summarizes the detection performance of our CD and the 5 baseline methods against 12 backdoor attacks. We test the performance on both the training and test sets (against the same poisoning rate of 5%). The AUROC for each detection method against each attack is averaged across all the DNN models. The detailed results for each dataset, attack, and model are provided in Appendix C, which are consistent with Table 1. Compared with existing detection methods ABL, AC,

Frequency, STRIP, and SS, our CD-based detection method achieves a new state-of-the-art performance consistently across almost all 12 backdoor attacks, 6 model architectures, and 3 datasets. Particularly, in training time detection, our CD-L method achieves an average AUROC of 96.45% against all the 12 attacks across the 3 datasets, which is 11% higher than the best baseline method Frequency. Such a stable performance of CD-L verifies the generalisability of our model and confirms our hypothesis about the characteristic of backdoor attacks: their CPs are smaller.

Table 1: The detection AUROC (%) of our CD method and the baselines against 12 backdoor attacks (poisoning rate 5%) on the *training/test* set. The results are averaged across the 6 models (VGG-16, RN-18, PARN-101, MobileV2, GoogLeNet, and EfficientNet-b0). The best results are in **bold**.

| Dataset | Attack | ABL | AC | Frequency | STRIP | SS | CD-L | CD-F |
|---|---|---|---|---|---|---|---|---|
| CIFAR10 | BadNets | 85.64/- | 77.57/74.63 | 92.32/91.59 | **97.89/97.66** | 62.89/45.50 | 94.03/94.72 | 88.89/89.88 |
| | Blend | 88.17/- | 76.23/65.93 | 80.67/79.40 | 84.55/83.02 | 51.63/40.52 | **93.47/93.44** | 92.30/92.41 |
| | CL | 90.86/- | 70.06/25.68 | **98.85**/91.59 | 97.27/**96.04** | 40.78/39.02 | 98.75/85.31 | 93.48/80.31 |
| | DFST | **89.10**/- | 80.45/86.97 | 87.62/87.34 | 58.08/58.51 | 56.34/40.69 | 88.96/**89.80** | 82.54/82.68 |
| | Dynamic | 87.97/- | 77.83/77.07 | 97.82/97.58 | 91.49/89.75 | 66.49/50.91 | **97.97/97.85** | 94.89/94.76 |
| | FC | 86.61/- | 83.99/88.74 | 98.65/98.11 | 79.84/76.97 | 63.62/64.62 | **99.17/98.22** | 94.46/95.12 |
| | SIG | **97.42**/- | 84.40/56.91 | 62.95/56.46 | 81.68/57.44 | 58.90/52.70 | 96.91/90.90 | 96.09/**93.17** |
| | Smooth | 79.53/- | 82.11/76.48 | 51.32/47.84 | 58.52/55.81 | 70.24/51.14 | **91.09/89.03** | 82.05/81.91 |
| | Nashville | 76.12/- | 89.26/76.11 | 70.53/67.71 | 51.62/48.30 | 80.48/60.62 | **98.10/97.34** | 95.28/94.26 |
| | Trojan | 85.96/- | 69.59/71.58 | 93.82/93.36 | 91.85/92.14 | 59.18/45.04 | **96.91/96.72** | 91.16/91.88 |
| | WaNet | 56.66/- | 70.96/69.86 | **96.31/96.65** | 84.98/84.64 | 71.59/57.27 | 95.69/96.08 | 86.60/88.43 |
| GTSRB | BadNets | 67.78/- | 98.21/72.79 | - | 57.26/59.59 | 69.97/72.86 | 99.28/99.14 | **99.59/99.66** |
| ImageNet | BadNets | 83.40/- | 95.75/**100.00** | - | 96.05/95.84 | 99.73/9.20 | **100.00/100.00** | 100.00/100.00 |
| | ISSBA | 96.99/- | **100.00**/80.29 | - | 70.37/68.73 | 42.22/56.31 | **100.00/99.99** | 99.97/99.89 |
| **Average** | - | 83.61/- | 82.60/73.21 | 84.62/82.51 | 78.61/75.96 | 63.83/49.58 | **96.45/94.90** | 92.66/91.74 |

Out of the 5 baselines, SS has the lowest performance. This is because SS requires backdoor labels for detection. However, here the backdoor labels are unknown to the defender. For SS to be effective, it may need to be used in conjunction with backdoor label detection methods. AC is the best baseline and is fairly stable across different attacks. This means that the deep features of backdoor samples are indeed anomalies in the deep feature space. Both AC and SS are deep feature-based detection methods, one may also observe from the full result in Appendix C that the performance of AC and SS against the same attack varies across different models. STRIP is quite effective against BadNets and, CL but performs poorly on DFST, Nashville, and Smooth attacks, which are all style transfer-based attacks. This suggests that the superimposition used by STRIP is vulnerable to stylized triggers. Note that our CD-L or CD-F is not always the best, for example, on BadNets, DFST, and SIG attacks, however, its performances are comparable to the best. For all detection methods, their test-time performance is relatively worse than their training-time performance, but the trends are consistent. SS demonstrates the largest performance drop, while our two CD methods, Frequency, and STRIP, experience the least drop. Our CD-L method achieves an average AUROC of 94.90% on the poisoned test sets, outperforming the best baseline Frequency by more than 12%.

## 4.2 ROBUST DETECTION UNDER DIFFERENT POISONING RATES

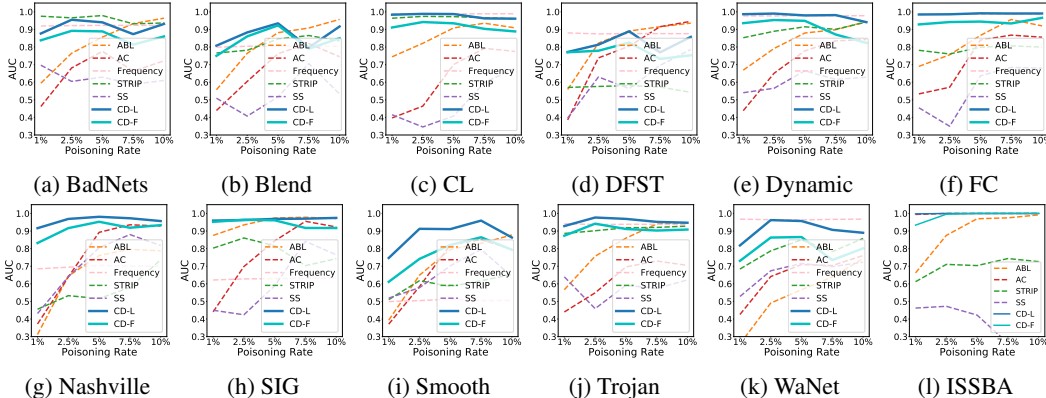

(a) BadNets  (b) Blend  (c) CL  (d) DFST  (e) Dynamic  (f) FC

(g) Nashville  (h) SIG  (i) Smooth  (j) Trojan  (k) WaNet  (l) ISSBA

Figure 3: The detection performance (AUROC) on CIFAR-10 training set under different poisoning rates (1%, 2.5%, 5%, 7.5%, 10%). Results are averaged across all 5 models following the same settings as in Table 1.

Here, we consider more challenging settings with varying poisoning rates [1%, 2.5%, 5%, 7.5%, 10%]. All other experimental settings are the same as in Section 4.1. Note that, under 1% poisoning rate, most of the attacks can still achieve an ASR ≥ 90%. The results on the training sets are presented in Figure 3. Similar results on the test sets can be found in Appendix C. It is evident that our method can achieve good detection performance robustly under different poisoning rates, except for a slight performance drop on Bend, DFST, Smooth, and WaNet under high poisoning rates > 5%. It can also be observed that ABL, AC, and SS are quite sensitive to the poisoning rate, but STRIP is comparably better. The Frequency is more stable under different poisoning rates. Only our methods and AC can robustly detect the more advanced attack ISSBA.

## 4.3 IMPROVING BACKDOOR MITIGATION

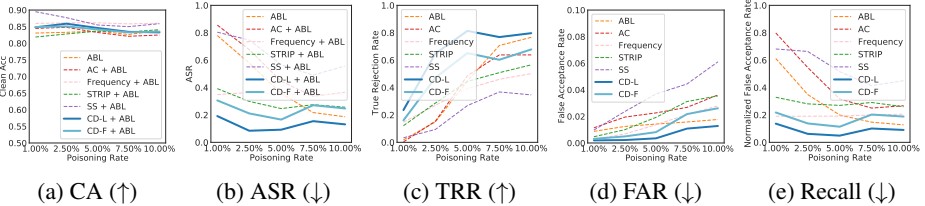

(a) CA (↑)  (b) ASR (↓)  (c) TRR (↑)  (d) FAR (↓)  (e) Recall (↓)

Figure 4: (a): Clean accuracy (CA) after mitigation; (b): Attack success rate (ASR) after mitigation; (c): True rejection rate (TRR); (d): False acceptance rate (FAR); (e) Recall measured on $\mathcal{D}'_c$ ($|\boldsymbol{x}_{bd} \in \mathcal{D}'_c|/|\mathcal{D}_b|$). All the experiments are averaged across all 5 models and 11 attacks on CIFAR-10 following the same setting as in Section 4.1.

Next, we demonstrate that effective backdoor sample detection can help improve backdoor mitigation methods. We follow one of the state-of-the-art mitigation methods ABL (Li et al., 2021a) to unlearn triggers from backdoored models using the detected backdoor subset $\mathcal{D}'_b$, and the clean subset $\mathcal{D}'_c$. ABL maximizes the model's loss on $\mathcal{D}'_b$ to remove the trigger while minimizing the loss on $\mathcal{D}'_c$ to maintain high clean accuracy. Based on the $L_1$ norm of the mask, here we select 2.5% samples with lower $L_1$ norm into $\mathcal{D}'_b$, and 70% samples with higher $L_1$ norm into $\mathcal{D}'_c$. We replace the backdoor isolation (detection) strategy in ABL with our two CD methods and denote the new methods as CD-L + ABL and CD-F + ABL, respectively. We run the experiments on all the attacks and models evaluated in Section 4.1. Detailed settings can be found in Appendix B.4.

The results are summarized in Figure 4. As can be observed, our CD-L method achieves the highest TRR and the lowest FAR, which both indicate good detection performance. Higher TRR means more backdoor samples are detected into $\mathcal{D}'_b$, facilitating more effective trigger unlearning. On the other hand, lower FAR means there are fewer backdoor samples left in $\mathcal{D}'_c$, which improves clean accuracy. Therefore, after mitigation, our CD-L method achieves the lowest ASR which is followed by our CD-F method. Note that the defender could also retrain the model on $\mathcal{D}'_c$, and in this case, FAR is the poisoning rate in $\mathcal{D}'_c$, i.e., lower is better. The above results indicate that accurate backdoor sample detection can provide substantial improvement to backdoor mitigation.

## 5 A CASE STUDY ON BIAS DETECTION

In this section, we show the connections between dataset bias and poisoning backdoor attacks. It has been shown that curated datasets may contain potential biases (Torralba & Efros, 2011), which will be memorized by DNN models trained on such datasets. It has also been found that bias may be strongly associated with a particular class. For example, gender is associated with occupation (Bolukbasi et al., 2016), person names Wang et al. (2022) or facial attributes (Zhang et al., 2018; Tartaglione et al., 2021). On the other hand, DNNs tend to learn more of the "easier" correlations (potential biases) in the dataset and this usually happens at an earlier training stage, as captured by the sample-wise training loss (Nam et al., 2020; Du et al., 2021). Such a phenomenon was also observed and leveraged in ABL (Li et al., 2021a) to isolate and mitigate backdoor attacks. These connections motivate us to apply our CD method to detect potential biases in a dataset. Here, we consider a real-world face dataset CelebA (Liu et al., 2015), which contains 40 binary facial attributes (the full training set is denoted as $D$). We train a classifier $f_i$ for each facial attribute $i$, based on a shared feature extractor (ResNet-18). The detailed setup is described in Appendix B.8.

We apply CD-L to select 0.5% samples with the lowest $\|\boldsymbol{m}\|_1$ for each classifier $f_i$ into a subset $D_i'$. We define the bias score as the distribution shift of attribute $j$ from $D$ to $D_i'$:

$$s(j, i) = (P_{ij} - P_j)/max(P_j, 1 - P_j), \tag{5}$$

where, $P_{ij} = |D_{ij}'|/|D_i'|$ is the percentage of samples in $D_i'$ that have attribute $j$, $P_j = |D_j|/|D|$ is the percentage of samples in the full training set that have attribute $j$. $s(j, i)$ measures to what degree attribute $j$ is predictive of attribute $i$ (according to $f_i$). A positive/negative $s(j, i)$ value means that (positive/negative) attribute $j$ is predictive of attribute $i$, closer to +1/-1 means more predictive. The $s(j, i)$ score allows us to identify the most predictive attributes to an attribute of interest, and potential biases if the absolute score is exceptionally high.

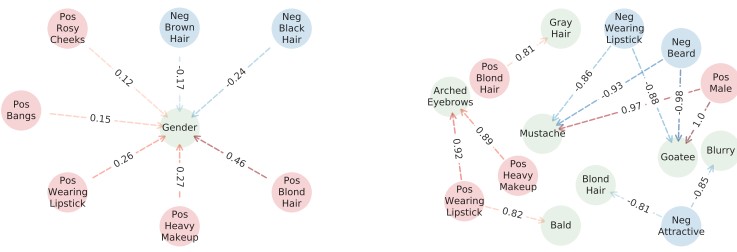

(a) Predictive attributes of *gender* attribute   (b) All highly correlated attributes

Figure 5: Potential biases detected from the CelebA dataset using our CD-L method. Each node in the graph is an attribute, with the links pointing (with score $s(j, i)$) from a highly predictive attribute to the attribute of interest (green nodes). Red/blue nodes and links represent positive/negative predictive attributes. Darker color edge indicates larger absolute $s(j, i)$ score.

As shown in Figure 5a, applying CD-L on the gender classifier, we identify the top-2 most predictive attributes of *gender*: *blond hair* and *heavy makeup*. This matches with one well-known bias in CelebA, i.e., *gender* is mostly determined by these two attributes (Nam et al., 2020; Tartaglione et al., 2021). Furthermore, our method also reveals two potential negative biases: *brown hair* and *black hair*, meaning that gender is hard to predict when these two hair colors appear. It also indicates that blond hair is mutually exclusive with other hair colors. Figure 5b further shows all the highly correlated (either positive or negative) attributes identify from CelebA, where the absolute $s(j, i)$ scores are all greater than 0.8. As suggested by Zhang et al. (2018), defining ground-truth biases is subject to different human interpretations. Several connections revealed in Figure 5b are closely related to known human stereotypes, such as males with different kinds of beards, or makeup with arched eyebrows. CD highlights the similarity between backdoor sample detection and bias detection. They both trigger over-easy predictions based on only a tiny amount of information perceived from the input. This means that even if a dataset does not contain backdoored samples, low $|\boldsymbol{m}|_1$ samples should be carefully examined as they may have biases. Our method demonstrates the possibility to detect potential biases in real-world datasets, a challenging problem in fair machine learning.

## 6   CONCLUSION

In this paper, we proposed a novel method, *Cognitive Distillation* (CD), that extracts the minimal pattern responsible for the model's prediction. With CD and the distilled patterns, we reveal that backdoored models make backdoor predictions only based on a sparse pattern of the input, regardless of the large trigger patterns used by different attacks. And the location of the pattern reveals the core part of the trigger. This allows us to build simple but effective backdoor sample detectors based on the learned input masks. We empirically show, with 6 DNN architectures, 3 datasets, and 12 advanced backdoor attacks that our proposed detection method is both effective and robust (under different poising rates). Our CD methods can also help detect potential biases in real-world datasets. The self-supervised nature of CD makes it a generic technique for investigating the underlying inference mechanism of different types of DNNs.

ACKNOWLEDGMENTS

Xingjun Ma is in part supported by the National Key R&D Program of China (Grant No. 2021ZD0112804), the National Natural Science Foundation of China (Grant No. 62276067), and the Science and Technology Commission of Shanghai Municipality (Grant No. 22511106102). Sarah Erfani is in part supported by Australian Research Council (ARC) Discovery Early Career Researcher Award (DECRA) DE220100680. This research was undertaken using the LIEF HPC-GPGPU Facility hosted at the University of Melbourne. This Facility was established with the assistance of LIEF Grant LE170100200. The authors would also like to thank Yige Li for sharing several of the backdoor triggers used in the experiments.

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

## A    DIFFERENCE OF CD TO EXISTING METHODS

**Difference with Model Interpretation Methods.** Our CD can provide a certain degree of explanation about how the model makes its predictions, thus related to model interpretation methods. Gradient-based interpretation methods generate saliency maps (Simonyan et al., 2013), such as deconvolution (Zeiler & Fergus, 2014) and guided backpropagation (Springenberg et al., 2014; Mahendran & Vedaldi, 2016) to help interpret how the model makes predictions given an input image. Perturbation-based methods (Dabkowski & Gal, 2017; Petsiuk et al., 2018; Fong et al., 2019; Yang et al., 2021; Zhao et al., 2021) are a generalized version of saliency maps that integrate several rounds of backpropagation to learn an explanation (Fong & Vedaldi, 2017). Meaningful Perturbations (MP) (Fong & Vedaldi, 2017) generate interpretations via deletion. It searches for a perturbed input that maximizes the score for a specific class. LIME (Ribeiro et al., 2016) is one of the foundational works for interpretation. It trains an additional locally faithful interpretable model to explain the target model. E.g., the faithful model can be a linear model that locally behaves the same as the target model. Similar to LIME, CD also emphasizes faithfulness, but it does not involve an additional model. Another set of methods is Class Activation Mapping (CAM) (Zhou et al., 2016) and its variants (Selvaraju et al., 2017; Chattopadhay et al., 2018). CAM-based methods use the gradient of the target concept (label) through a specific layer and generate a coarse localization map highlighting important regions.

**Technical Comparison.** The difference between our CD and closely-related methods, including LIME (Ribeiro et al., 2016), Meaningful Perturbation (MP) (Fong & Vedaldi, 2017), and Neural Cleanse (NC) (Wang et al., 2019), are summarized in Table 2. Different from LIME, MP, and NC, our CD method is self-supervised, that is, the distillation process does not rely on the class labels. And CD searches for the most determining input pattern of the model output, for whichever class it predicts.

Table 2: The objective functions of LIME, Meaningful Perturbation (MP), Neural Cleanse (NC), and our Cognitive Distillation (CD). $\mathcal{L}_{CE}$ is the cross-entropy loss.

| Method | Optimization Objective |
|---|---|
| LIME | $\arg\min\limits_{g \in G} \mathcal{L}(f, g, \pi_x) + \Omega(g)$ |
| NC | $\arg\min\limits_{\boldsymbol{m}, \boldsymbol{\Delta}, t \in 1 \dots K} \mathcal{L}_{CE}(f((1 - \boldsymbol{m}_t) \odot \boldsymbol{x} + \boldsymbol{m}_t \odot \boldsymbol{\Delta}_t), y_t) + \alpha\|\boldsymbol{m}_t\|_1$ |
| MP | $\arg\min\limits_{\boldsymbol{m}} \mathcal{L}_{CE}(f(\boldsymbol{x} \odot \boldsymbol{m} + (1 - \boldsymbol{m}) \odot \phi), y_t)) + \alpha\|1 - \boldsymbol{m}\|_1$ |
| **CD (ours)** | $\arg\min\limits_{\boldsymbol{m}} \|f(\boldsymbol{x}) - f(\boldsymbol{x} \odot \boldsymbol{m} + (1 - \boldsymbol{m}) \odot \delta)\|_1 + \alpha\|\boldsymbol{m}\|_1 + \beta TV(\boldsymbol{m})$ |

**Detailed Difference with LIME.** LIME aims to find an additional interpretable model $g$, with $\Omega(g)$ regularizing its complexity. $\mathcal{L}$ enforces the local fidelity that $g$ behaves faithfully to $f$ (DNN) around $\boldsymbol{x}$ (using the mask to perturb around $\boldsymbol{x}$). When applying LIME on images, ridge regression is selected as $g$, and one requires a targeted class $y_t$ intended to explain. The output is a coarse heatmap defined by fixed super-pixels. However, our method is different from LIME, and there is no additional model $g$ or targeted class $y_t$ involved. Also, our generated masks are fine-grained with the most important pixels for the model's prediction.

**Detailed Difference with NC.** NC optimizes for both a mask and the trigger pattern. The optimization outputs a mask $\boldsymbol{m}_t$ and a trigger pattern $\Delta_t$ for a class $y_t$. The objective of NC is to disentangle a perturbed input image $\boldsymbol{x}'$ into a class-wise universal trigger pattern $\Delta_t$ with a corresponding mask $\boldsymbol{m}_t$ and residual image $(1 - \boldsymbol{m}_t) \odot \boldsymbol{x}$. For $\boldsymbol{x} \in \mathcal{D}$, the perturbed input $\boldsymbol{x}'$ can make the model predict the target class $y_t$. NC can be used for backdoored label detection, trigger synthesis (recovery) and backdoor model detection. It measures the $L_1$ norm of the mask $\boldsymbol{m}_t$ for all classes and the Median Absolute Deviation for detecting backdoored models and labels, rather than backdoor samples. Also note that existing works (Liu et al., 2019; Nguyen & Tran, 2020; Li et al., 2021c; Cheng et al., 2020) have demonstrated that advanced attacks can still evade NC's detection. As discussed in NC (Wang et al., 2019), it is also not efficient for models/datasets with a large number of classes/labels.

Different from NC, our CD aims to find the pattern for each sample in the dataset rather than a class of samples. Both CD's output and main optimization functions are different from NC: 1) CD outputs sample-specific masks and patterns, 2) its objective is the *self-supervised* absolute difference

between $f(\boldsymbol{x})$ and $f(\boldsymbol{x}')$, rather than the *supervised* cross-entropy loss ($\mathcal{L}_{CE}$). Moreover, CD's computational complexity does not increase with the number of classes. NC (Wang et al., 2019) and its related works (Guo et al., 2019; Xu et al., 2021) find that the backdoor class requires less perturbation to perturbed into than other classes. Unlike NC, our results are from an input-specific perspective that useful features of backdoored input samples contain fewer pixels than clean samples.

**Detailed Difference with MP.** MP aims to find a perturbation mask that can explain a target class $y_t$ through $\mathcal{L}_{CE}$. The target class is also known as a *concept* that needs to be explained by the model. The deleted perturbation operator $\phi$ includes constant, noise, and blur. MP aims to find deletion regions that are maximally informative (for $y_t$), while CD aims to find regions that can be preserved for the prediction (and does not require $y_t$). For the removed pixel, CD uses a random color value $\delta$ in each optimization step, while the $\phi$ is the same operation applied in every step.

MP is more similar to NC than our CD, since both of them use CE as the main objective. The backdoor class label of the image must be known in advance for backdoor sample detection.

In a nutshell, MP asks the model *what can be removed to make a cat image not like a 'cat'?*, however, our CD asks *what is the minimal sufficient pattern the model needs to make its prediction?*

## B EXPERIMENTAL SETTINGS

All experiments are run with NVIDIA Tesla P100/V100/A100 GPUs with PyTorch implementations. Code is available at `https://github.com/HanxunH/CognitiveDistillation`. For CIFAR10 experiments, we train for 60 epochs with SGD optimizer, weight decay $5 \times 10^{-4}$, initial learning rate 0.1 decay by 0.1 at the 45th epoch. For GTSRB, we use the same set up with CIFAR10, except for 40 epochs and decay at the 35th epcoh. For the ImageNet subset, we train 200 epochs with the same learning rate settings and weight decays, except the learning rate adopts a cosine decay without restart (Loshchilov & Hutter, 2017).

### B.1 DATASETS

The datasets used in our experiments in Section 4.1 are summarized in Table 3.

Table 3: Statistics of the datasets used in our experiments

| Dataset | No. Classes | Input Size | No. Training Images |
|---|---|---|---|
| CIFAR10 (Krizhevsky et al., 2009) | 10 | 32×32×3 | 50,000 |
| GTSRB (Houben et al., 2013) | 43 | 32×32×3 | 39,209 |
| ImageNet (Deng et al., 2009) subset* | 200 | 224×224×3 | 100,000 |

\* The split is consistent with Li et al. (2021c).

### B.2 ATTACK CONFIGURATIONS

Table 4: Configurations and implementation details of the backdoor attacks.

| Attacks | Trigger Size | Trigger Pattern | Trigger Type | Strategy |
|---|---|---|---|---|
| BadNets (Gu et al., 2017) | Patch | Grid | Class-wise | Pixel modification |
| Blend (Chen et al., 2017) | Full image | Hello Kitty | Class-wise | Blend in |
| CL (Turner et al., 2018) | Full image | Patch + Adv Noise | Class-wise | Optimizations |
| DFST (Cheng et al., 2020) | Full image | Style transfer | Sample-specific | GAN |
| Dynamic (Nguyen & Tran, 2020) | Few pixels | Mask generator | Sample-specific | Mask Generator |
| FC (Shafahi et al., 2018) | Full image | Optimized Noise | Sample-specific | Optimization |
| SIG (Barni et al., 2019) | Full image | Periodical signal | Class-wise | Blend in |
| Trojan (Liu et al., 2018b) | Patch | Watermark | Class-wise | Pixel modification |
| WaNet (Nguyen & Tran, 2021) | Full image | Invisible noise | Sample-specific | Wrapping function |
| Nashville (Liu et al., 2019) | Full image | Style transfer | Sample-specific | Instagram filter |
| Smooth (Zeng et al., 2021) | Full image | Style transfer | Sample-specific | Smooth frequency space |
| ISSBA (Li et al., 2021c) | Full image | Mask Generator | Sample-specific | Encoder |

The detailed configurations of the 12 backdoor attacks considered in our experiments are summarized in Table 4. The class-wise pattern means applying the same pattern on all images of a particular

class, while the sample-specific pattern means that each sample has a specific pattern. We have verified their attack success rate (ASR) and clean accuracy (CA) on the test sets, and all the attacks are effective, as shown in Appendix C. Visualizations of the backdoor samples are in Appendix D. For both Dynamic (Nguyen & Tran, 2020) and WaNet (Nguyen & Tran, 2021), we use their trigger but do not manipulate the training objective, which is consistent with the threat model. For WaNet, we find its original hyperparameters are not strong enough to generate effective triggers in this threat model. We thus increase the uniform grid size to $32 \times 32$ to improve its ASR.

## B.3 DETECTION CONFIGURATIONS

Table 5: Detection configurations of backdoor sample detection methods. For all baseline methods, we followed their official open-sourced implementations.

| Method | Detection confidence score |
|---|---|
| ABL (Li et al., 2021a) | Training loss of first 20 epochs |
| AC (Chen et al., 2018) | Silhouette scores |
| Frequency (Zeng et al., 2021) | Detector's output probability |
| STRIP (Gao et al., 2019) | Entropy |
| SS (Tran et al., 2018) | Top right singular vector |

Configurations of the sample detection methods used in the experiments are summarized in Table 5. ABL is a hybrid method of both backdoor sample detection and backdoor mitigation. Here we only use its sample detection part, which was referred to as backdoor sample isolation in the paper (Li et al., 2021a). We use the sample-specific training loss averaged over the first 20 epochs of model training as the confidence score. For AC, SS and STRIP, they are specifically designed for poison sample detection, and we use their original settings. For SS, we use the top right singular vector of the SVD decomposition as the confidence score. In our threat model, it assumes that the defender does not have prior knowledge of which samples are clean, which is a more realistic in real-world scenarios. For STRIP, we use all training data as the sampling pool for superimposing as the defender does not have any clean samples. Interestingly, we find this does not dramatically affect SRTIP's performance.

For Frequency defense, we followed the strategy that converts images into frequency space using a Discrete Cosine Transform. Given that the defender may not have access to clean data or know the type of trigger, we train the backdoor detector on the GTSRB dataset and a wide range of generated triggers that does not assume any prior knowledge of the type of trigger or location. We exclude the Frequency defense method from evaluations on GTSRB and ImageNet, since the detector is trained on GTSRB, and ImageNet has different resolutions that require an additional detector. We use the detector's output probability as the confidence score.

For test-time detection, except for STRIP, other baseline methods cannot be directly applied. This is because both AC and SS require ground truth label $y$, for which we replace it with the model's prediction. It should have minimal impact on the detection performance if the prediction is accurate, should degrade the performance otherwise. We exclude ABL from our test-time detection experiments, as it requires training statistics, and there is no intuitive solution to accommodate this requirement. We assume all the methods can calculate the mean score on 1% of the clean training set and use 1 standard deviation lower than the mean as the threshold.

**CD.** The $\beta$ is set to 10. The $\alpha$ is set to 0.01/0.001 for models trained on CIFAR10 and GTSRB, 0.001/0.0001 for the ImageNet subset model, for CD-L/CD-F, respectively. We use Adam optimizer (Kingma & Ba, 2014) with initial learning rate 0.1, $\beta_1$=0.1, $\beta_2$=0.1, and a total of 100 steps to learn the input mask. We use the scaled tanh function to ensure the mask is between [0, 1].

## B.4 BACKDOOR MITIGATION

Following previous work (Li et al., 2021a), the defender can take a step further to remove the backdoor via a process of anti-backdoor finetuning. I.e., finetuning the backdoored model to maximizing its loss on $\mathcal{D}'_b$ while minimizing its loss on $\mathcal{D}'_c$, as following:

$$\arg\min_{\theta} \mathbb{E}_{(\boldsymbol{x}_c, y_c) \sim \mathcal{D}'_c} \mathcal{L}(f_\theta(\boldsymbol{x}_c), y_c) + \arg\max_{\theta} \mathbb{E}_{(\boldsymbol{x}_b, y_b) \sim \mathcal{D}'_b} \mathcal{L}(f_\theta(\boldsymbol{x}_b), y_b). \tag{6}$$

We use the method described in Algorithm 1. We set $p_b$ to 2.5% and $p_c$ to 70%, and optimize for 5 epochs with learning rate set to $5 \times 10^{-4}$.

---

**Algorithm 1** Unlearning and Fine-tuning

---

1: **Input:** $f_\theta$, $\mathcal{D}$, percentage $p_b$ for $\mathcal{D}'_b$, percentage $p_c$ for $\mathcal{D}'_c$, Scores $\|m\|_1$, Epochs $E$
2: $\mathcal{D} = \text{argsort}(\mathcal{D}, s)$
3: $\mathcal{D}'_b = $ select $p_b$ samples with lower $s$ from $\mathcal{D}$          ▷ detected as backdoor samples
4: $\mathcal{D}'_c = $ select $p_c$ samples with higher $s$ from $\mathcal{D}$       ▷ detected as clean samples
5: **for** $e$ **to** $E$ **do**
6:      optimize $\arg\min_\theta \mathbb{E}_{(\boldsymbol{x},y)\sim\mathcal{D}'_c} \mathcal{L}(f_\theta(\boldsymbol{x}), y)$
7:      optimize $\arg\max_\theta \mathbb{E}_{(\boldsymbol{x},y)\sim\mathcal{D}'_b} \mathcal{L}(f_\theta(\boldsymbol{x}), y)$
8: **end for**
9: **Output:** $f_\theta$

---

## B.5    ANALYSIS OF THE REGULARIZATION TERMS

The goal of our CD is to find the most determining pattern from the input image of the model's prediction via a learnable mask. Intuitively, one can use following objective function:

$$\arg\min_{\boldsymbol{m}} \|f_\theta(\boldsymbol{x}) - f_\theta(\boldsymbol{x} \odot \boldsymbol{m})\|_1 + \alpha\|(\boldsymbol{m})\|_1, \qquad (7)$$

where, $\boldsymbol{x} \odot \boldsymbol{m}$ is the perturbed input and the unimportant pixels are replaced with 0. The above objective may not be ideal as there may exist black/darker pixels with values very close to 0, which will be surely removed by the $\|\boldsymbol{m}\|_1$ regularization. However, in clean images, black/darker pixels may also be important. As shown in Figure 6, the optimized mask only considers the white pixels of the trigger are necessary for the model's prediction, with the darker pixels are completely ignored. For the clean samples, darker pixels of the main objects ( "horse" or "peacock") are also important.

In order to address the above issue, we replace the unimportant pixels with a random noise vector $\delta$ sampled at each optimization step, using the following objective function:

$$\arg\min_{\boldsymbol{m}} \|f_\theta(\boldsymbol{x}) - f_\theta(\boldsymbol{x} \odot \boldsymbol{m} + (1 - \boldsymbol{m}) \odot \delta)\|_1 + \alpha\|(\boldsymbol{m})\|_1, \qquad (8)$$

where, random noise $\delta$ is applied to all spatial dimensions. This objective function will force the mask to consider darker pixels as equally important for the model's prediction as white pixels. As shown in Figure 7, the optimized mask now can reveal the black pixels of the backdoor trigger. For clean samples, darker pixels of the main objects ("horse" and "peacock") can now also be considered as important by our CD.

In order to make the extracted mask to be smooth and more interpretable, we add one more regularization term to CD's objective:

$$\arg\min_{\boldsymbol{m}} \|f_\theta(\boldsymbol{x}) - f_\theta(\boldsymbol{x} \odot \boldsymbol{m} + (1 - \boldsymbol{m}) \odot \delta)\|_1 + \alpha\|(\boldsymbol{m})\|_1 + \beta TV(\boldsymbol{m}), \qquad (9)$$

where, the $TV$ is the total variation loss and $\beta$ balances the $TV$ with other objectives. This ensures the model to consider the surrounding pixels of an important pixel as similarly important. As shown in Figure 8, the learned mask with the $TV$ regularization can outline the shape of the main object.

## B.6    ABLATION STUDY OF THE HYPERPARAMETERS

In this section, we perform a comprehensive ablation study of the hyperparameters used by CD, in the context of backdoor detection performance.

**Sensitivity to $\alpha$.** The $\alpha$ hyperparameter controls the number of pixels that can be kept in the distilled pattern. If too many or too few pixels are removed from the pattern, then it would be difficult to distinguish between the backdoor and clean samples. $\alpha$ should be set in a way that the extracted pattern mainly focuses on the main object in the image. In practice, the defender may generate patterns for a small subset of samples to examine the effect of the pattern. As shown in Figure 9a, $\alpha = 0.01$ shows the extracted pattern focuses on the main object of the image, and the

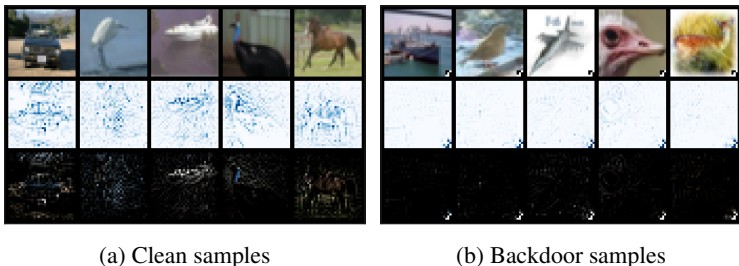

| (a) Clean samples | (b) Backdoor samples |

Figure 6: First row: examples of clean and backdoor images (BadNets) randomly selected from the CIFAR10 training set. Second row: the corresponding learned masks $m$ using Equation 7. Third row: the distilled patterns $x \odot m$ for the corresponding masks and images.

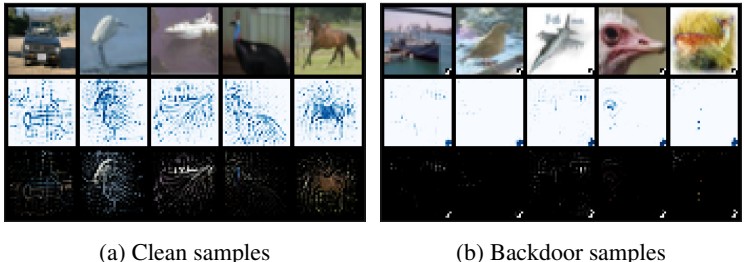

| (a) Clean samples | (b) Backdoor samples |

Figure 7: Learned masks $m$ using Equation 8.

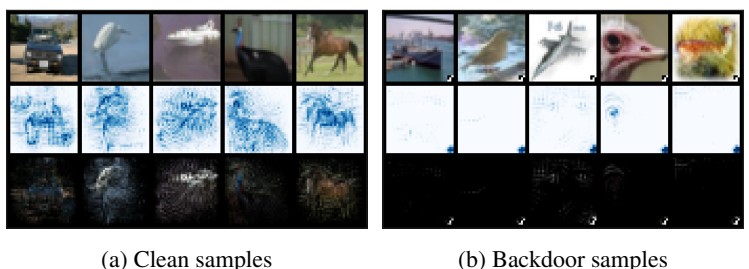

| (a) Clean samples | (b) Backdoor samples |

Figure 8: Learned masks $m$ using Equation 9.

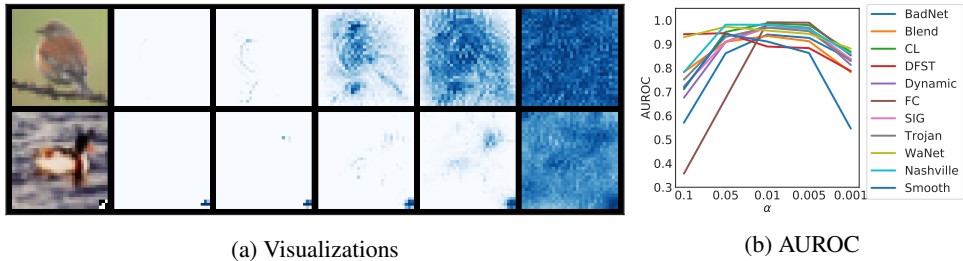

| (a) Visualizations | (b) AUROC |

Figure 9: Varying $\alpha$ in [0.1, 0.05, 0.01, 0005, 0.001] for our CD. (a) Visualizations of the learned mask for clean (top row) and backdoor (bottom row) samples. (b) The detection performance of AUROC on the training set for each attack with 5% poisoning rate. Results are averaged across models.

mask removes other pixels. From Figure 9b, it can also be observed that the detection performance is stable from 0.05 to 0.005 against all attacks.

**Sensitivity to $\beta$.** The $\beta$ hyperparameter controls how smooth is the learned mask. Large $\beta$ tends to produce smoothed mask, while small $\beta$ will generate a sparse mask. As shown in Figure 10a, a

smoothed mask could produce a more interpretable pattern that matches with human understanding. As shown in Figure 10b, $\beta$ does not significantly affect the backdoor detection performance.

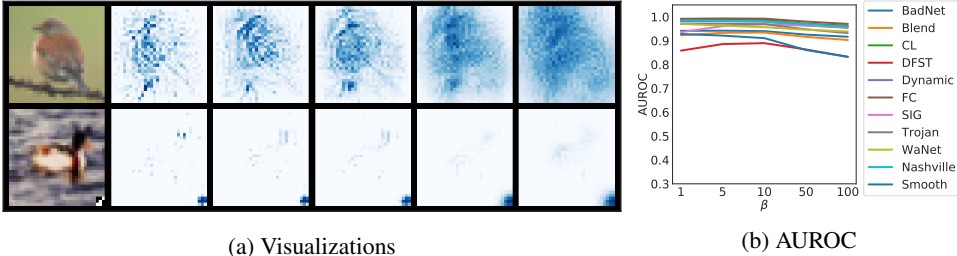

(a) Visualizations           (b) AUROC

Figure 10: Varying $\beta$ in [1, 5, 10, 50, 100] for CD. (a) Visualizations of the learned mask for clean (top row) and backdoor (bottom row) samples. (b) The detection performance of AUROC on the training set for each attack with poisoning rate 5%. Results are averaged across models.

### B.7 WHITE-BOX ADAPTIVE ATTACKS

In this section, we evaluate our CD-based detection method against white-box adaptive attacks, i.e., the attacker is aware of our detection strategy and attempts to evade our detection with more adaptive strategies . Overall, we find that, in order to evade our detection, the adversary has to sacrifice one of the two crucial components of backdoor attacks: attack success rate or stealthiness. The detailed evaluations are as follows.

**Increasing trigger visibility.** Our CD uses the $L_1$ norm of the distilled trigger patterns as the detection metric. As shown in Figure 1, the attacked models by existing attacks do not fully use the entire trigger pattern for the backdoor predictions. And our detection leverages this characteristic to detect backdoor samples. Intuitively, the adversary can intentionally increase the size of the trigger pattern perceived by the model to evade our detection by increasing either 1) the size of the patch trigger, or 2) the visibility of the blending trigger (e.g., reducing the transparency). Here, we test this type of adaptive attacks based on BadNets and Blend attacks, as both methods can be easily adapted to different trigger sizes and degrees of visibility.

**Decreasing Attack Success Rate.** The adversarial may alternative use a weaker trigger to evade our detection, i.e., trigger that blends well into the clean image by being not so adversarial. In this case, the trigger will not be (so) responsible for the model's prediction, which could evade our detection to some extent. This can be achieved by decreasing either 1) the poisoning rate, or 2) visibility of the trigger. Here, we take BadNets, SIG, and Nashville as three example attacks to demonstrate this adaptive strategy. We also evaluate the performance of the Blend attack by reducing its transparency.

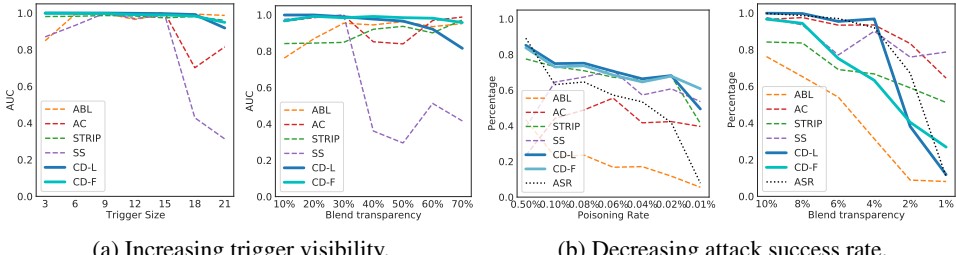

(a) Increasing trigger visibility.           (b) Decreasing attack success rate.

Figure 11: (a): Increasing the trigger size of BadNets and the transparency of Blend. (b): Decreasing the poisoning rate to lower ASRs (results averaged across BadNest, SIG and Nashville attacks) and the transparency of Blend. All experiments are evaluated on CIFAR10 with ResNet-18. The results (AUROCs) are calculated on the training set.

As shown in Figure 11a, if the size or the visibility of the trigger increases, it could reduce the detection performance of our CD. However, this will greatly impact the stealthiness of the attack, as the trigger patterns visualized in Figure 12. The results show that the trigger size needs to be as large

as the main content in the image to evade our detection. Figure 11b shows that the adversary could sacrifice certain ASR to evade our detection, and other detection methods. Our CD is comparably more robust against poisoning rate decrease, but is less robust against the adaptive Blend attack with reduced transparency if it becomes lower than 2%. Both STRIP and our CD are based on input-output responses, and if the attacks are barely successful, then the detection performance could be affected. Nevertheless, in either case, the effectiveness (stealthiness or ASR) of the attack will be greatly impacted.

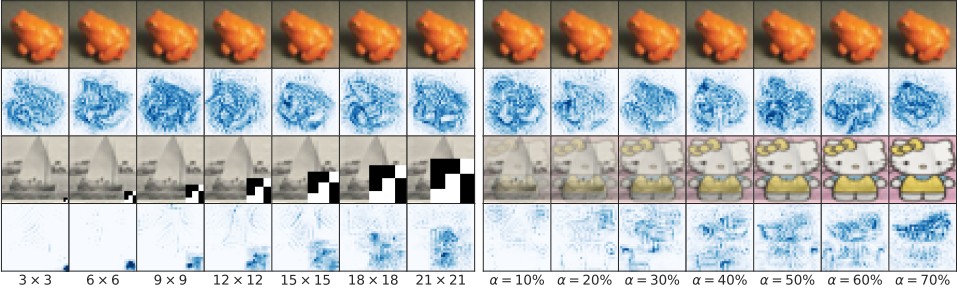

| 3 × 3 | 6 × 6 | 9 × 9 | 12 × 12 | 15 × 15 | 18 × 18 | 21 × 21 | $\alpha = 10\%$ | $\alpha = 20\%$ | $\alpha = 30\%$ | $\alpha = 40\%$ | $\alpha = 50\%$ | $\alpha = 60\%$ | $\alpha = 70\%$ |

(a) Increasing the trigger size of BadNets     (b) Increasing the trigger transparency of Blend

Figure 12: First row: examples of clean images from the CIFAR10 training set. Second row: the corresponding learned masks of clean images extracted for each model trained under different trigger sizes and visibility. Third row: examples of the backdoor image with different trigger sizes and visibility. Forth row: the corresponding learned masks of backdoor images extracted for each model trained under different trigger sizes and visibility.

**Training-objective adaptive attack.**

We investigate if CD could be evaded if the training objective is manipulated. Note that, in our threat model, the attackers do not have access to the training procedure and thus cannot alter the training objective. Here, we go beyond this assumption and test whether our CD defense, in a pure white-box setting, can stand such adaptive attacks. Our CD technique can be viewed as a generalized version of the saliency map methods, and the mask is generated through several rounds of backpropagation. CD uses $|\boldsymbol{m}|_1$ to classify if the sample is backdoored. To evade the detection, the attacker may add a regularization term to the original training objective that forces the model to pay attention to all pixels of the trigger pattern. Here, we realize this attack by the following input gradient regularization, which is designed to force the model to produce uniform attention over the backdoor samples. Formally, it is defined as:

$$\mathbb{E}_{(\boldsymbol{x},y)\sim\mathcal{D}_b}\left[\mathcal{L}(f(\boldsymbol{x}),y) + \lambda\|\|\boldsymbol{\nabla}_x\mathcal{L}(f(\boldsymbol{x}),y) - \gamma\|_1\right] + \mathbb{E}_{(\boldsymbol{x},y)\sim\mathcal{D}_c}\left[\mathcal{L}(f(\boldsymbol{x}),y)\right], \qquad (10)$$

where $\mathcal{D}_b$ is the backdoored subset, $\mathcal{D}_c$ is the clean subset, $\mathcal{L}$ is the cross-entropy loss, $\gamma$ is the targeted value of the input gradient and the $\lambda$ balances the strength of the regularization.

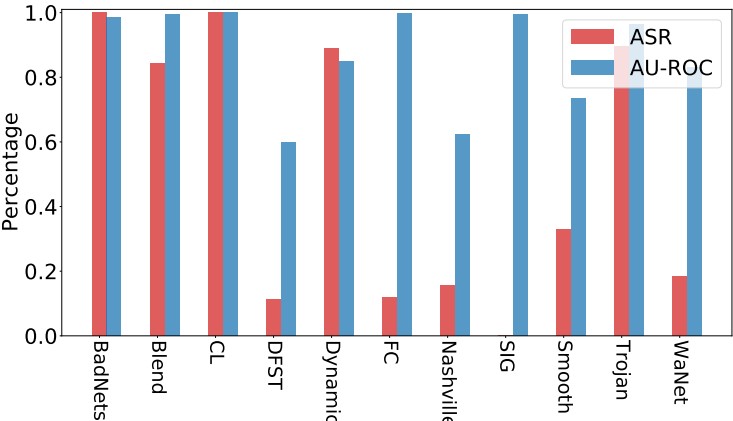

Figure 13: ResNet-18 trained on CIFAR-10 with input gradient regularization.

We use double backpropagation (Drucker & Le Cun, 1991) to solve the above adaptive objective, with $\gamma = 0.0001$ and $\lambda = 10.0$. All other settings are the same as our main experiments. The results are shown in Figure 13 to 15. It can be observed that this adaptive attack strategy can not evade our CD detection. For patch-based triggers (BadNets, CL, and Trojan), Dynamic, and Blend attacks, the AUROC is above 0.8. For the style transfer-like triggers (DFST, Nashville, and Smooth), the adaptive attack strategy can indeed cause CD to generate larger $|\boldsymbol{m}|_1$ for backdoored samples. However, this tends to sacrifice its ASR ($\leq 40\%$). Interestingly, for FC, SIG, and WaNet, the CD can detect backdoor samples even if the ASR is low. The above results indicate that our defense is not easily broken even if the training objective is adaptively manipulated.

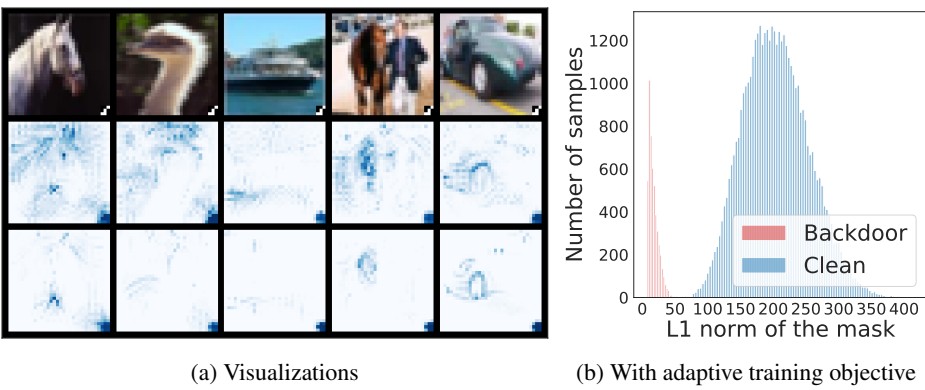

(a) Visualizations        (b) With adaptive training objective

Figure 14: (a) First row: input (BadNets attack) images. Second row: masks generated based on a regular backdoored model. Third row: masks generated based on the adaptively backdoored model. (b) The distribution of $|\boldsymbol{m}|_1$ for both clean and backdoor samples. This experiment is conducted on CIFAR-10 with BadNets (Gu et al., 2017). The ASR of the adaptive backdoor model is **100%**.

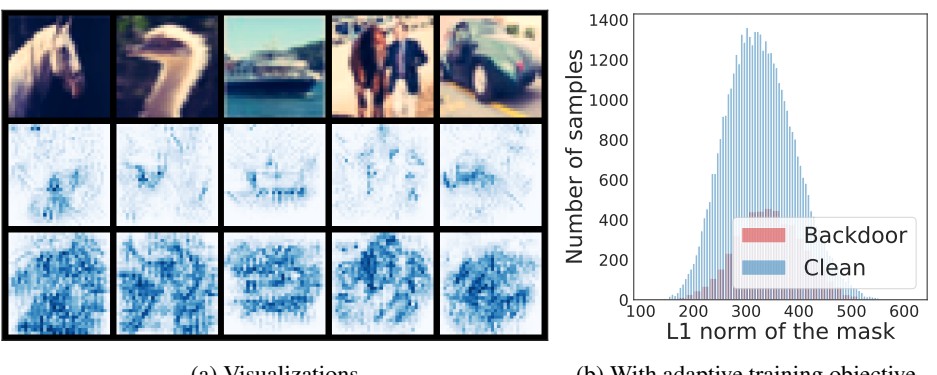

(a) Visualizations        (b) With adaptive training objective

Figure 15: (a) First row: input (Nashville attack) images. Second row: masks generated based on a regular backdoored model. Third row: masks generated based on the adaptively backdoored model. (b) The distribution of $|\boldsymbol{m}|_1$ for both clean and backdoor samples. The experiment is conducted on CIFAR-10 with Nashville (Liu et al., 2019). The ASR of the adaptive backdoor model is **15.6%**.

## B.8    BIAS DETECTION

**Experiment setting.** For all the 40 binary facial attributes, we use a ResNet-18 (He et al., 2016a) as the feature extractor and attach a classifier head for each attribute. All classifiers share the same feature extractor and are trained with the cross-entropy loss. We train the model for 10 epochs using SGD, with weight decay $5 \times 10^{-4}$. For CD, we set $\alpha$ to 0.001, $\beta$ to 1000 and optimize the mask for 300 steps with learning rate 0.075. Figure 16 visualizes the learned masks using our CD-L method. It shows that the masks can locate the area for each facial attribute that matches with human understanding, suggesting that CD-L provides reasonable explanations for the model's prediction.

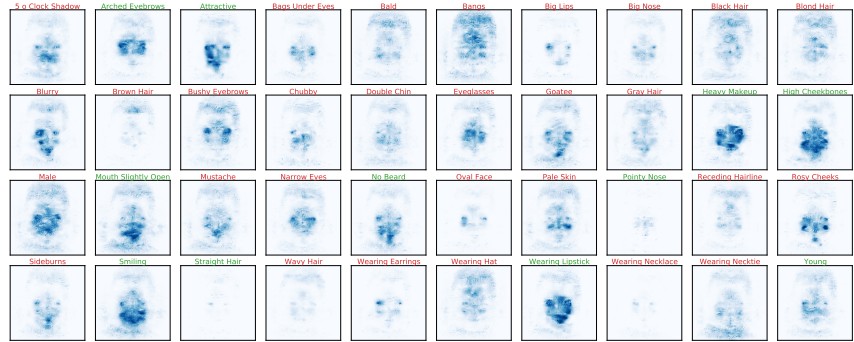

Figure 16: Visualization of the learned masks using CD-L in the bias detection experiments. Positive facial attributes are indicated as green and negative as red.

## C ADDITIONAL EXPERIMENT RESULTS

In this section, we show additional results, including the detection performance under varying poisoning rates on the test set in Figure 17, and detailed results that are summarized in Table 1 in the main paper. Detailed results for training and test set detection using area under the ROC curve (AUROC) are in Table 6 and Table 8, using area under the precision recall curve (AUPRC) are in Table 7 and Table 9, respectively. The result of using TRP/FPR with one standard deviation ($\gamma = 1$) lower than the mean calculated on 1% of the training subset as the threshold is in Table 10.

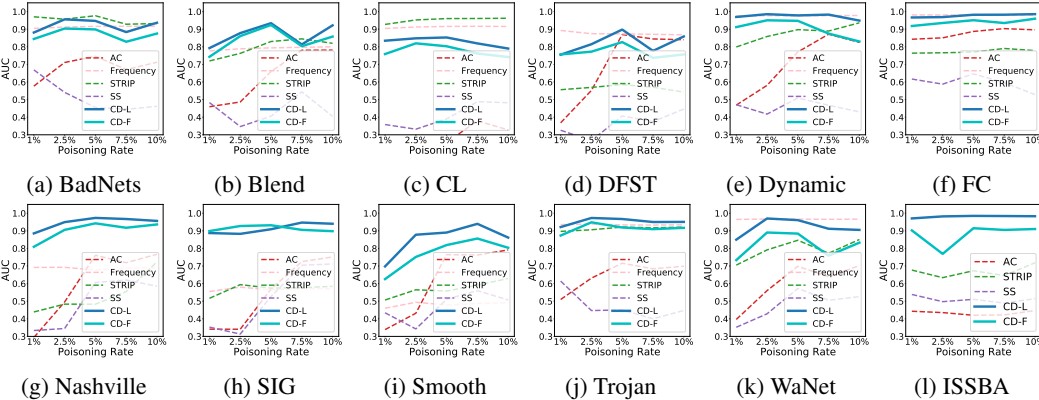

Figure 17: The detection performance (AUROC) on test set with different attacks under different poisoning rates [1%, 2.5%, 5%, 7.5%, 10%]. Results are averaged across different models.

Table 6: The detection performance on training set measured by AUROC ((%)) for our CD method and the baselines against 12 backdoor attacks. The poisoning rate is 5% for all attacks. The Clean Accuracy (CA) is evaluated on clean test set. The best detection results are in **bold**.

| Dataset | Attack | Model | CA | ASR | ABL | AC | STRIP | SS | CD-L | CD-F |
|---|---|---|---|---|---|---|---|---|---|---|
| CIFAR10 | BadNets | VGG-16 | 89.05 | 100.00 | 88.31 | 64.41 | 94.82 | 21.41 | **99.98** | 99.96 |
| | | RN-18 | 87.60 | 100.00 | 85.87 | 99.97 | 98.68 | 99.48 | **100.00** | 99.99 |
| | | MobileV2 | 84.19 | 100.00 | 72.70 | 91.38 | 98.15 | 67.42 | **99.73** | 94.88 |
| | | PARN-101 | 89.77 | 100.00 | 83.58 | 45.05 | 98.65 | **99.58** | 74.10 | 67.69 |
| | | GoogLeNet | 92.84 | 100.00 | 97.74 | 87.05 | **99.13** | 26.54 | 96.34 | 81.92 |
| | Blend | VGG-16 | 89.26 | 99.60 | 90.50 | 65.05 | 76.37 | 22.58 | 97.46 | **97.76** |
| | | RN-18 | 87.77 | 100.00 | 88.78 | 78.67 | 78.67 | 96.20 | **99.92** | 96.20 |
| | | MobileV2 | 84.81 | 99.40 | 85.69 | 46.56 | 88.08 | 59.43 | **98.40** | 84.91 |
| | | PARN-101 | 89.66 | 99.8 0 | 91.54 | 75.69 | 90.76 | 41.33 | **97.89** | 94.03 |
| | | GoogLeNet | 92.39 | 99.20 | 84.32 | **95.04** | 88.86 | 38.61 | 73.66 | 88.62 |
| | CL | VGG-16 | 89.00 | 98.40 | 89.88 | 80.19 | 92.76 | 46.52 | **99.99** | 99.80 |
| | | RN-18 | 87.33 | 98.40 | 89.95 | 81.91 | 97.91 | 39.15 | **99.97** | 98.44 |
| | | MobileV2 | 85.15 | 90.00 | 82.95 | 59.28 | 98.02 | 34.46 | **98.90** | 89.82 |
| | | PARN-101 | 89.65 | 93.60 | 94.75 | 38.35 | **98.90** | 44.45 | 95.60 | 88.08 |
| | | GoogLeNet | 92.69 | 53.60 | 96.77 | 90.57 | 98.75 | 39.31 | **99.25** | 91.26 |
| | DFST | VGG-16 | 88.09 | 100 | 81.83 | 37.90 | 61.93 | 6.73 | 92.53 | **95.59** |
| | | RN-18 | 87.34 | 100.00 | 96.31 | 97.12 | 57.89 | **98.73** | 98.42 | 95.52 |
| | | MobileV2 | 84.67 | 99.80 | 80.41 | **84.47** | 61.07 | 38.40 | 81.15 | 68.65 |
| | | PARN101 | 89.37 | 99.60 | **94.91** | 83.83 | 57.56 | 42.42 | 76.85 | 85.46 |
| | | GoogLeNet | 92.90 | 100.00 | 92.04 | **98.93** | 51.95 | 95.39 | 95.83 | 67.49 |
| | Dynamic | VGG-16 | 88.94 | 100.00 | 93.12 | 54.18 | 80.94 | 48.16 | **99.86** | 99.14 |
| | | RN-18 | 87.93 | 100.00 | 96.83 | 95.55 | 84.98 | 93.44 | **99.99** | 99.42 |
| | | MobileV2 | 84.85 | 99.40 | 85.15 | 82.24 | 97.40 | 67.34 | **99.09** | 92.65 |
| | | PARN101 | 89.41 | 99.60 | 81.39 | 59.74 | **95.43** | 80.03 | 90.97 | 85.25 |
| | | GoogLeNet | 92.99 | 100.00 | 83.35 | 97.43 | 98.71 | 43.47 | **99.95** | 98.01 |
| | FC | VGG-16 | 89.06 | 99.00 | 92.25 | 75.76 | 69.17 | 27.41 | **96.72** | 77.32 |
| | | RN-18 | 88.17 | 99.60 | 85.40 | 98.82 | 82.63 | 99.84 | **99.98** | 99.68 |
| | | MobileV2 | 86.50 | 99.90 | 90.43 | 71.58 | 83.08 | 58.40 | **99.36** | 98.31 |
| | | PARN101 | 89.65 | 99.80 | 83.75 | 74.19 | 85.17 | 32.93 | **99.78** | 97.99 |
| | | GoogLeNet | 92.14 | 100.00 | 81.23 | 99.62 | 79.16 | 99.54 | **99.99** | 99.02 |
| | Nashville | VGG-16 | 88.89 | 98.60 | 82.06 | 69.08 | 49.29 | 39.70 | 94.13 | **95.30** |
| | | RN-18 | 87.86 | 99.00 | 54.02 | 92.08 | 53.43 | 95.01 | **98.85** | 92.52 |
| | | MobileV2 | 85.24 | 99.40 | 74.62 | 97.24 | 49.03 | 88.94 | **99.69** | 98.87 |
| | | PARN101 | 89.31 | 98.60 | 86.91 | 89.08 | 49.50 | 81.46 | 97.97 | 94.52 |
| | | GoogLeNet | 92.79 | 99.80 | 82.98 | 98.79 | 56.86 | 97.28 | **99.87** | 95.18 |
| | SIG | VGG-16 | 89.22 | 95.60 | **97.57** | 44.36 | 78.26 | 39.42 | 92.22 | 87.92 |
| | | RN-18 | 86.71 | 99.80 | 99.22 | 99.90 | 81.89 | 99.96 | **99.99** | 99.95 |
| | | MobileV2 | 84.45 | 90.80 | **98.10** | 80.94 | 80.13 | 28.82 | 95.54 | 95.44 |
| | | PARN101 | 88.92 | 99.80 | 99.12 | **99.64** | 85.90 | 89.71 | 98.80 | 99.23 |
| | | GoogLeNet | 92.80 | 94.20 | 93.09 | 97.17 | 82.23 | 36.62 | **97.97** | 97.92 |
| | Smooth | VGG-16 | 88.11 | 96.20 | **87.52** | 47.68 | 65.59 | 28.99 | 86.37 | 78.00 |
| | | RN-18 | 87.45 | 95.20 | 79.36 | 95.60 | 70.13 | **99.82** | 97.78 | 86.50 |
| | | MobileV2 | 85.80 | 96.00 | 76.06 | 81.10 | 65.31 | 30.72 | **93.08** | 82.48 |
| | | PARN101 | 88.57 | 96.20 | 67.70 | 91.98 | 52.64 | **93.01** | 83.18 | 73.38 |
| | | GoogLeNet | 92.75 | 96.00 | 87.03 | 94.18 | 38.93 | **98.66** | 95.05 | 89.91 |
| | Trojan | VGG-16 | 88.47 | 98.20 | 88.42 | 47.16 | 88.90 | 47.80 | **99.11** | 99.00 |
| | | RN-18 | 87.63 | 98.40 | 81.71 | 98.74 | 91.88 | 93.76 | **99.85** | 98.81 |
| | | MobileV2 | 85.03 | 98.00 | 80.81 | 67.62 | 90.14 | 43.66 | **98.22** | 88.97 |
| | | PARN101 | 88.30 | 98.20 | 87.31 | 56.20 | **94.12** | 80.96 | 91.17 | 83.60 |
| | | GoogLeNet | 92.94 | 98.20 | 91.54 | 78.24 | 94.21 | 29.71 | **96.20** | 85.44 |
| | WaNet | VGG-16 | 88.53 | 97.80 | 58.73 | 44.48 | 90.24 | 53.94 | **96.73** | 92.21 |
| | | RN-18 | 87.51 | 98.40 | 51.42 | 91.41 | 80.45 | **99.75** | 95.71 | 85.04 |
| | | MobileV2 | 85.75 | 96.40 | 70.96 | 68.66 | 83.39 | 41.85 | **99.06** | 90.36 |
| | | PARN101 | 88.46 | 98.40 | 37.66 | 52.72 | 76.00 | 67.61 | **87.34** | 70.32 |
| | | GoogLeNet | 92.70 | 99.80 | 64.52 | 97.51 | 94.79 | 94.81 | **99.62** | 95.05 |
| GTSRB | BadNets | VGG-16 | 97.58 | 100.00 | 49.39 | 99.48 | 72.69 | 70.01 | **100.00** | 99.99 |
| | | RN-18 | 98.12 | 100.00 | 87.73 | 99.45 | 68.38 | 65.09 | 97.35 | **100.00** |
| | | MobileV2 | 97.32 | 100.00 | 64.27 | 99.43 | 58.63 | 81.98 | **100.00** | 98.71 |
| | | PARN101 | 96.68 | 100.00 | 39.94 | 92.95 | 39.14 | 55.06 | **100.00** | 99.73 |
| | | GoogLeNet | 99.20 | 100.00 | 97.56 | **99.74** | 47.47 | 77.72 | 99.06 | 99.50 |
| ImageNet Subset | BadNet | RN-18 | 58.13 | 100.00 | 94.05 | 97.87 | 98.94 | 99.83 | **100.00** | **100.00** |
| | | EfficientNet-b0 | 66.41 | 100.00 | 72.75 | 93.63 | 93.16 | 99.64 | **100.00** | **100.00** |
| | ISSBA | RN-18 | 57.11 | 100.00 | 97.69 | **100.00** | 67.49 | 32.22 | **100.00** | 99.98 |
| | | EfficientNet-b0 | 62.19 | 100.00 | 96.29 | **100.00** | 73.24 | 52.22 | **100.00** | 99.97 |
| Average | - | - | 87.66 | 97.91 | 83.00 | 81.17 | 78.17 | 63.16 | **96.12** | 91.98 |

Table 7: The detection performance on training set measured by AUPRC ((%)) for our CD method and the baselines against 12 backdoor attacks. The poisoning rate is 5% for all attacks. The Clean Accuracy (CA) is evaluated on clean test set. The best detection results are in **bold**.

| Dataset | Attack | Model | CA | ASR | ABL | AC | STRIP | SS | CD-L | CD-F |
|---|---|---|---|---|---|---|---|---|---|---|
| CIFAR10 | BadNets | VGG-16 | 89.05 | 100.00 | 17.41 | 6.43 | 37.09 | 2.93 | **99.92** | 97.20 |
| | | RN-18 | 87.60 | 100.00 | 14.35 | 99.77 | 64.36 | 89.49 | **100.00** | 99.81 |
| | | MobileV2 | 84.19 | 100.00 | 7.89 | 68.43 | 62.33 | 7.45 | **97.18** | 68.04 |
| | | PARN101 | 89.77 | 100.00 | 12.86 | 4.10 | 64.66 | **91.72** | 12.78 | 9.92 |
| | | GoogLeNet | 92.84 | 100.00 | **78.14** | 16.22 | 75.62 | 3.11 | 67.80 | 25.47 |
| | Blend | VGG-16 | 89.26 | 99.60 | 45.00 | 6.49 | 14.44 | 3.00 | 76.04 | **82.06** |
| | | RN-18 | 87.77 | 100.00 | 32.27 | 93.35 | 17.72 | 61.49 | **98.74** | 49.28 |
| | | MobileV2 | 84.81 | 99.40 | 47.90 | 4.34 | 30.44 | 5.72 | **85.46** | 23.75 |
| | | PARN101 | 89.66 | 99.80 | 54.31 | 11.30 | 34.87 | 3.85 | **75.42** | 42.76 |
| | | GoogLeNet | 92.39 | 99.20 | 34.60 | **74.17** | 35.26 | 3.67 | 9.30 | 24.67 |
| | CL | VGG-16 | 89.00 | 98.40 | 41.29 | 20.14 | 46.03 | 4.20 | **99.90** | 96.60 |
| | | RN-18 | 87.33 | 98.40 | 23.63 | 19.10 | 73.55 | 3.71 | **99.64** | 87.95 |
| | | MobileV2 | 85.15 | 90.00 | 13.39 | 6.90 | 79.41 | 3.43 | **89.09** | 52.14 |
| | | PARN101 | 89.65 | 93.60 | 51.73 | 3.65 | **82.35** | 4.51 | 63.32 | 40.62 |
| | | GoogLeNet | 92.69 | 53.60 | 64.60 | 53.27 | 81.22 | 3.72 | **91.90** | 57.01 |
| | DFST | VGG-16 | 88.09 | 100.00 | 15.58 | 3.75 | 5.91 | 2.63 | 41.53 | **60.99** |
| | | RN-18 | 87.34 | 100.00 | 81.82 | **93.03** | 5.36 | 78.89 | 90.55 | 60.94 |
| | | MobileV2 | 84.67 | 99.80 | 12.64 | **30.17** | 5.75 | 3.65 | 22.44 | 9.20 |
| | | PARN101 | 89.37 | 99.60 | **84.82** | 16.47 | 5.31 | 3.90 | 11.32 | 22.62 |
| | | GoogLeNet | 92.90 | 100.00 | 51.09 | **97.20** | 4.68 | 40.15 | 70.85 | 7.51 |
| | Dynamic | VGG-16 | 88.94 | 100.00 | 55.75 | 5.13 | 36.35 | 4.45 | **98.58** | 91.11 |
| | | RN-18 | 87.93 | 100.00 | 87.78 | 86.97 | 37.91 | 51.34 | **99.86** | 94.84 |
| | | MobileV2 | 84.85 | 99.40 | 35.18 | 53.09 | 85.67 | 8.97 | **92.97** | 50.54 |
| | | PARN101 | 89.41 | 99.60 | 17.91 | 7.90 | **72.73** | 40.78 | 53.67 | 33.49 |
| | | GoogLeNet | 92.99 | 100.00 | 36.64 | 87.68 | 89.10 | 4.06 | **99.40** | 84.75 |
| | FC | VGG-16 | 89.06 | 99.00 | 55.87 | 9.80 | 17.21 | 3.13 | **58.01** | 11.02 |
| | | RN-18 | 88.17 | 99.60 | 40.13 | 94.81 | 37.46 | 94.53 | **99.65** | 96.26 |
| | | MobileV2 | 86.50 | 99.90 | 48.76 | 23.53 | 51.90 | 6.51 | **93.05** | 83.82 |
| | | PARN101 | 89.65 | 99.80 | 32.19 | 32.82 | 46.07 | 3.36 | **96.78** | 81.46 |
| | | GoogLeNet | 92.14 | 100.00 | 22.46 | 98.03 | 40.20 | 82.94 | **99.91** | 91.15 |
| | Nashville | VGG-16 | 88.89 | 98.60 | 33.66 | 7.58 | 5.37 | 3.73 | **72.67** | 58.28 |
| | | RN-18 | 87.86 | 99.00 | 5.16 | 76.86 | 6.06 | 69.72 | **91.51** | 51.21 |
| | | MobileV2 | 85.24 | 99.40 | 12.60 | 87.52 | 5.33 | 17.65 | **96.81** | 92.22 |
| | | PARN101 | 89.31 | 98.60 | 56.19 | 49.90 | 5.51 | 13.32 | **87.04** | 64.47 |
| | | GoogLeNet | 92.79 | 99.80 | 23.62 | 97.54 | 6.22 | 48.44 | **99.16** | 66.11 |
| | SIG | VGG-16 | 89.22 | 95.60 | **80.92** | 4.08 | 20.05 | 3.74 | 33.51 | 20.59 |
| | | RN-18 | 86.71 | 99.80 | 94.79 | 99.66 | 21.97 | 98.50 | **99.89** | 99.21 |
| | | MobileV2 | 84.45 | 90.80 | **82.08** | 45.09 | 16.94 | 3.20 | 63.97 | 64.92 |
| | | PARN101 | 88.92 | 99.80 | 93.78 | **98.24** | 28.00 | 18.81 | 79.23 | 89.72 |
| | | GoogLeNet | 92.80 | 94.20 | 43.54 | **88.30** | 24.57 | 3.57 | 83.04 | 81.09 |
| | Smooth | VGG-16 | 88.11 | 96.20 | **42.97** | 4.54 | 14.12 | 3.19 | 33.98 | 15.82 |
| | | RN-18 | 87.45 | 95.20 | 33.60 | 91.22 | 15.59 | **96.23** | 84.55 | 27.94 |
| | | MobileV2 | 85.80 | 96.00 | 11.91 | 29.42 | 12.18 | 3.26 | **59.04** | 24.23 |
| | | PARN101 | 88.57 | 96.20 | 17.90 | **58.45** | 6.55 | 32.05 | 27.37 | 11.08 |
| | | GoogLeNet | 92.75 | 96.00 | 30.68 | **85.53** | 4.01 | 73.62 | 73.38 | 42.63 |
| | Trojan | VGG-16 | 88.47 | 98.20 | 23.53 | 4.31 | 41.11 | 4.36 | **98.01** | 96.73 |
| | | RN-18 | 87.63 | 98.40 | 11.72 | 95.56 | 59.55 | 42.87 | **99.44** | 97.04 |
| | | MobileV2 | 85.03 | 98.00 | 12.18 | 9.43 | 55.72 | 4.04 | **92.14** | 49.50 |
| | | PARN101 | 88.30 | 98.80 | 23.15 | 5.32 | **66.44** | 23.42 | 46.28 | 24.85 |
| | | GoogLeNet | 92.94 | 98.20 | 60.93 | 10.75 | **74.06** | 3.22 | 66.94 | 34.57 |
| | WaNet | VGG-16 | 88.53 | 97.80 | 5.57 | 4.10 | 33.90 | 5.43 | **73.02** | 71.88 |
| | | RN-18 | 87.51 | 98.40 | 4.60 | 76.57 | 18.69 | **93.87** | 37.00 | 31.11 |
| | | MobileV2 | 85.75 | 96.40 | 8.15 | 9.60 | 20.11 | 3.87 | **91.45** | 50.62 |
| | | PARN101 | 88.46 | 98.40 | 3.64 | 4.91 | 15.81 | 12.32 | **22.28** | 8.72 |
| | | GoogLeNet | 92.70 | 99.80 | 6.47 | 93.69 | 51.52 | 38.40 | **94.16** | 60.21 |
| GTSRB | BadNets | VGG-16 | 97.58 | 100.00 | 4.50 | 99.50 | 10.33 | 7.40 | **100.00** | 99.82 |
| | | RN-18 | 98.12 | 100.00 | 38.06 | 99.30 | 8.27 | 6.35 | 44.90 | **99.92** |
| | | MobileV2 | 97.32 | 100.00 | 6.65 | 99.33 | 5.91 | 11.44 | **100.00** | 92.65 |
| | | PARN101 | 96.68 | 100.00 | 3.79 | 72.98 | 3.69 | 5.17 | **99.99** | 94.59 |
| | | GoogLeNet | 99.20 | 100.00 | 60.97 | **99.44** | 4.61 | 9.66 | 67.97 | 85.39 |
| ImageNet Subset | BadNet | RN-18 | 58.13 | 100.00 | 27.81 | 96.71 | 77.01 | 92.13 | **100.00** | **100.00** |
| | | EfficientNet-b0 | 66.41 | 100.00 | 8.00 | 91.76 | 33.11 | 83.44 | **100.00** | **100.00** |
| | ISSBA | RN-18 | 57.11 | 100.00 | 83.02 | **100.00** | 9.17 | 3.17 | 99.98 | 99.77 |
| | | EfficientNet-b0 | 62.19 | 100.00 | 56.83 | **100.00** | 14.49 | 4.47 | 99.98 | 99.48 |
| Average | - | - | 87.66 | 97.91 | 36.35 | 51.96 | 32.88 | 25.52 | **76.78** | 61.6 |

Table 8: The detection performance on test set measured by AUROC ((%)) for our CD method and the baselines against 12 backdoor attacks. The poisoning rate is 5% for all attacks. The Clean Accuracy (CA) is evaluated on clean test set. The best detection results are in **bold**.

| Dataset | Attack | Model | CA | ASR | AC | STRIP | SS | CD-L | CD-F |
|---|---|---|---|---|---|---|---|---|---|
| CIFAR10 | BadNets | VGG-16 | 89.05 | 100.00 | 72.08 | 93.95 | 14.48 | **100.00** | 99.96 |
| | | RN-18 | 87.60 | 100.00 | 90.86 | 98.61 | 10.74 | **100.00** | 99.97 |
| | | MobileV2 | 84.19 | 100.00 | 91.52 | 98.15 | 72.02 | **99.79** | 94.70 |
| | | PARN101 | 89.77 | 100.00 | 36.43 | **98.54** | 97.85 | 77.63 | 71.90 |
| | | GoogLeNet | 92.84 | 100.00 | 82.29 | **99.05** | 32.43 | 96.18 | 82.91 |
| | Blend | VGG-16 | 89.26 | 99.60 | 59.55 | 74.25 | 18.34 | 96.53 | **96.96** |
| | | RN-18 | 87.77 | 100.00 | 97.34 | 78.02 | 58.43 | **99.77** | 96.17 |
| | | MobileV2 | 84.81 | 99.40 | 40.17 | 85.55 | 77.30 | **97.98** | 84.83 |
| | | PARN101 | 89.66 | 99.80 | 40.47 | 90.09 | 20.26 | **97.69** | 94.49 |
| | | GoogLeNet | 92.39 | 99.20 | **92.11** | 87.18 | 28.30 | 75.22 | 89.63 |
| | CL | VGG-16 | 89.00 | 98.40 | 18.78 | 93.52 | 44.32 | 93.08 | **96.05** |
| | | RN-18 | 87.33 | 98.40 | 29.54 | 96.95 | 22.26 | **97.48** | 91.70 |
| | | MobileV2 | 85.15 | 90.00 | 39.38 | **98.14** | 47.00 | 87.14 | 74.19 |
| | | PARN101 | 89.65 | 93.60 | 24.62 | **97.20** | 45.10 | 83.58 | 79.29 |
| | | GoogLeNet | 92.69 | 53.60 | 16.09 | **94.41** | 36.39 | 65.27 | 60.33 |
| | DFST | VGG-16 | 88.09 | 100.00 | 86.80 | 63.30 | 7.72 | 93.06 | **95.15** |
| | | RN-18 | 87.34 | 100.00 | 96.51 | 59.93 | 73.92 | **97.22** | 94.49 |
| | | MobileV2 | 84.67 | 99.80 | **83.46** | 59.90 | 24.04 | 82.25 | 69.12 |
| | | PARN101 | 89.37 | 99.60 | 69.32 | 57.72 | 16.17 | 80.81 | **87.26** |
| | | GoogLeNet | 92.90 | 100.00 | **98.78** | 51.71 | 81.61 | 95.66 | 67.37 |
| | Dynamic | VGG-16 | 88.94 | 100.00 | 60.53 | 77.44 | 33.08 | **98.93** | 98.68 |
| | | RN-18 | 87.93 | 100.00 | 93.78 | 82.74 | 49.19 | **99.97** | 99.35 |
| | | MobileV2 | 84.85 | 99.40 | 79.60 | 96.75 | 67.43 | **98.54** | 91.85 |
| | | PARN101 | 89.41 | 99.60 | 54.23 | **93.72** | 75.30 | 91.88 | 86.03 |
| | | GoogLeNet | 92.99 | 100.00 | 97.20 | 98.12 | 29.53 | **99.92** | 97.89 |
| | FC | VGG-16 | 89.06 | 99.00 | 87.52 | 70.44 | 83.47 | **92.79** | 82.23 |
| | | RN-18 | 88.17 | 99.60 | 94.93 | 83.07 | 58.01 | **99.68** | 99.08 |
| | | MobileV2 | 86.50 | 99.90 | 79.55 | 79.36 | 69.23 | **99.26** | 97.81 |
| | | PARN101 | 89.65 | 99.80 | 85.54 | 82.51 | 45.66 | **99.40** | 97.76 |
| | | GoogLeNet | 92.14 | 100.00 | 96.14 | 69.49 | 66.74 | **99.96** | 98.71 |
| | Nashville | VGG-16 | 88.89 | 98.60 | 50.02 | 45.40 | 24.16 | 92.05 | **92.26** |
| | | RN-18 | 87.86 | 99.00 | 87.50 | 53.26 | 69.54 | **97.08** | 91.20 |
| | | MobileV2 | 85.24 | 99.40 | 96.33 | 44.50 | 90.32 | **99.59** | 98.91 |
| | | PARN101 | 89.31 | 98.60 | 53.99 | 44.96 | 36.43 | **98.04** | 93.95 |
| | | GoogLeNet | 92.79 | 99.80 | 92.71 | 53.37 | 82.63 | **99.95** | 94.97 |
| | SIG | VGG-16 | 89.22 | 95.60 | 40.89 | 55.44 | 38.97 | 72.27 | **76.74** |
| | | RN-18 | 86.71 | 99.80 | 97.71 | 69.52 | 95.26 | **99.85** | 99.68 |
| | | MobileV2 | 84.45 | 90.80 | 43.10 | 47.35 | 43.47 | 91.42 | **93.88** |
| | | PARN101 | 88.92 | 99.80 | 48.44 | 67.77 | 49.20 | 96.41 | **98.35** |
| | | GoogLeNet | 92.80 | 94.20 | 54.41 | 47.12 | 36.61 | 94.57 | **97.21** |
| | Smooth | VGG-16 | 88.11 | 96.20 | 63.37 | 63.27 | 25.35 | **82.23** | 77.65 |
| | | RN-18 | 87.45 | 95.20 | 88.60 | 67.64 | 79.47 | **94.30** | 85.27 |
| | | MobileV2 | 85.80 | 96.00 | 78.83 | 62.87 | 25.76 | **92.48** | 81.56 |
| | | PARN101 | 88.57 | 96.20 | 61.75 | 48.10 | 32.85 | **82.60** | 75.77 |
| | | GoogLeNet | 92.75 | 96.00 | 89.87 | 37.15 | 92.25 | **93.52** | 89.31 |
| | Trojan | VGG-16 | 88.47 | 98.20 | 60.54 | 89.21 | 27.25 | **98.15** | 98.14 |
| | | RN-18 | 87.63 | 98.40 | 97.31 | 92.52 | 41.21 | **99.17** | 98.35 |
| | | MobileV2 | 85.03 | 98.00 | 71.84 | 90.85 | 44.33 | **97.79** | 89.50 |
| | | PARN101 | 88.30 | 98.80 | 58.61 | **93.94** | 76.13 | 92.83 | 86.57 |
| | | GoogLeNet | 92.94 | 98.20 | 69.61 | 94.19 | 36.26 | **95.65** | 86.84 |
| | WaNet | VGG-16 | 88.53 | 97.80 | 56.29 | 88.48 | 31.21 | **95.63** | 91.15 |
| | | RN-18 | 87.51 | 98.40 | 91.14 | 83.57 | 96.63 | **97.09** | 89.39 |
| | | MobileV2 | 85.75 | 96.40 | 60.15 | 79.14 | 47.02 | **98.49** | 88.55 |
| | | PARN101 | 88.46 | 98.40 | 43.49 | 77.36 | 33.32 | **89.55** | 77.55 |
| | | GoogLeNet | 92.70 | 99.80 | 98.24 | 94.66 | 78.19 | **99.65** | 95.51 |
| GTSRB | BadNets | VGG-16 | 97.58 | 100.00 | 59.50 | 76.42 | 77.76 | **100.00** | 100.00 |
| | | RN-18 | 98.12 | 100.00 | 83.77 | 70.27 | 76.22 | 96.98 | **100.00** |
| | | MobileV2 | 97.32 | 100.00 | 68.54 | 61.03 | 90.09 | **100.00** | 98.82 |
| | | PARN101 | 96.68 | 100.00 | 54.55 | 41.23 | 23.16 | **100.00** | 99.75 |
| | | GoogLeNet | 99.20 | 100.00 | 97.61 | 49.02 | 97.05 | 98.72 | **99.71** |
| ImageNet Subset | BadNets | RN-18 | 58.13 | 100.00 | **100.00** | 98.78 | 9.66 | **100.00** | 100.00 |
| | | EfficientNet-b0 | 66.41 | 100.00 | **100.00** | 92.90 | 8.74 | **100.00** | 100.00 |
| | ISSBA | RN-18 | 57.11 | 100.00 | 80.84 | 65.84 | 67.01 | **99.99** | 99.93 |
| | | EfficientNet-b0 | 62.19 | 100.00 | 79.74 | 71.62 | 45.60 | **99.99** | 99.85 |
| Average | - | - | 87.66 | 97.91 | 71.62 | 75.36 | 51.16 | **94.42** | 90.97 |

Table 9: The detection performance on test set measured by AUPRC ((%)) for our CD method and the baselines against 12 backdoor attacks. The poisoning rate is 5% for all attacks. The Clean Accuracy (CA) is evaluated on clean test set. The best detection results are in **bold**.

| Dataset | Attack | Model | CA | ASR | AC | STRIP | SS | CD-L | CD-F |
|---|---|---|---|---|---|---|---|---|---|
| CIFAR10 | BadNets | VGG-16 | 89.05 | 100.00 | 7.94 | 32.74 | 2.79 | **100.00** | 96.21 |
| | | RN-18 | 87.60 | 100.00 | 22.00 | 62.89 | 2.71 | **99.99** | 99.59 |
| | | MobileV2 | 84.19 | 100.00 | 66.77 | 59.68 | 9.10 | **97.40** | 67.86 |
| | | PARN101 | 89.77 | 100.00 | 3.64 | 61.09 | 72.07 | 13.84 | 10.78 |
| | | GoogLeNet | 92.84 | 100.00 | 12.18 | **72.08** | 3.37 | 66.15 | 26.45 |
| | Blend | VGG-16 | 89.26 | 99.60 | 5.56 | 12.38 | 2.86 | 75.24 | **83.41** |
| | | RN-18 | 87.77 | 100.00 | 91.62 | 16.62 | 5.85 | **97.69** | 48.83 |
| | | MobileV2 | 84.81 | 99.40 | 3.76 | 26.24 | 10.76 | **83.59** | 22.97 |
| | | PARN101 | 89.66 | 99.80 | 3.86 | 33.82 | 2.92 | **73.48** | 44.17 |
| | | GoogLeNet | 92.39 | 99.20 | **79.19** | 31.69 | 3.16 | 9.85 | 25.50 |
| | CL | VGG-16 | 89.00 | 98.40 | 2.93 | 31.18 | 4.64 | 82.88 | **87.57** |
| | | RN-18 | 87.33 | 98.40 | 3.28 | 48.85 | 2.96 | **81.95** | 51.87 |
| | | MobileV2 | 85.15 | 90.00 | 3.71 | **65.29** | 4.43 | 38.22 | 15.08 |
| | | PARN101 | 89.65 | 93.60 | 3.14 | **52.32** | 4.42 | 22.82 | 17.56 |
| | | GoogLeNet | 92.69 | 53.60 | 2.87 | **36.53** | 3.55 | 8.62 | 6.77 |
| | DFST | VGG-16 | 88.09 | 100.00 | 17.36 | 6.12 | 2.64 | 46.78 | **65.40** |
| | | RN-18 | 87.34 | 100.00 | **92.52** | 5.68 | 10.86 | 88.59 | 55.71 |
| | | MobileV2 | 84.67 | 99.80 | 19.19 | 5.58 | 3.01 | **23.62** | 9.50 |
| | | PARN101 | 89.37 | 99.60 | 7.57 | 5.37 | 2.82 | 13.49 | **24.14** |
| | | GoogLeNet | 92.90 | 100.00 | **97.65** | 4.66 | 13.04 | 69.85 | 7.57 |
| | Dynamic | VGG-16 | 88.94 | 100.00 | 6.06 | 30.20 | 3.38 | **96.67** | 88.25 |
| | | RN-18 | 87.93 | 100.00 | 83.15 | 30.62 | 4.52 | **99.67** | 93.27 |
| | | MobileV2 | 84.85 | 99.40 | 43.36 | 84.12 | 9.85 | **91.21** | 49.02 |
| | | PARN101 | 89.41 | 99.60 | 5.35 | **65.98** | 32.66 | 51.69 | 30.96 |
| | | GoogLeNet | 92.99 | 100.00 | 89.72 | 85.54 | 3.21 | **98.99** | 83.95 |
| | FC | VGG-16 | 89.06 | 99.00 | **80.25** | 25.39 | 21.69 | 60.63 | 36.64 |
| | | RN-18 | 88.17 | 99.60 | 90.39 | 46.22 | 10.27 | **98.59** | 96.10 |
| | | MobileV2 | 86.50 | 99.90 | 60.99 | 46.05 | 21.54 | **95.92** | 90.26 |
| | | PARN101 | 89.65 | 99.80 | 75.21 | 47.10 | 7.77 | **96.84** | 88.34 |
| | | GoogLeNet | 92.14 | 100.00 | 91.70 | 33.30 | 12.62 | **99.70** | 94.40 |
| | Nashville | VGG-16 | 88.89 | 98.60 | 4.55 | 4.32 | 3.13 | **69.53** | 54.46 |
| | | RN-18 | 87.86 | 99.00 | 73.57 | 5.11 | 14.11 | **86.07** | 47.59 |
| | | MobileV2 | 85.24 | 99.40 | 78.97 | 4.19 | 20.08 | **96.34** | 91.77 |
| | | PARN101 | 89.31 | 98.60 | 4.95 | 4.28 | 3.57 | **86.74** | 64.57 |
| | | GoogLeNet | 92.79 | 99.80 | 33.16 | 5.04 | 11.95 | **99.29** | 66.73 |
| | SIG | VGG-16 | 89.22 | 95.60 | 3.90 | 6.13 | 4.19 | 12.14 | **17.94** |
| | | RN-18 | 86.71 | 99.80 | 93.53 | 10.62 | 36.67 | **98.39** | 95.81 |
| | | MobileV2 | 84.45 | 90.80 | 3.97 | 4.51 | 4.12 | 44.23 | **47.16** |
| | | PARN101 | 88.92 | 99.80 | 4.58 | 9.66 | 4.44 | 59.63 | **80.38** |
| | | GoogLeNet | 92.80 | 94.20 | 4.92 | 4.79 | 3.56 | 61.32 | **72.31** |
| | Smooth | VGG-16 | 88.11 | 96.20 | 6.55 | 11.89 | 3.07 | **28.64** | 14.49 |
| | | RN-18 | 87.45 | 95.20 | **77.84** | 12.56 | 15.92 | 77.01 | 24.10 |
| | | MobileV2 | 85.80 | 96.00 | 31.21 | 10.00 | 3.06 | **58.04** | 22.50 |
| | | PARN101 | 88.57 | 96.20 | 6.31 | 5.35 | 3.35 | **25.82** | 11.36 |
| | | GoogLeNet | 92.75 | 96.00 | **78.54** | 3.82 | 38.34 | 71.78 | 44.15 |
| | Trojan | VGG-16 | 88.47 | 98.20 | 5.65 | 39.78 | 3.13 | 96.31 | **96.77** |
| | | RN-18 | 87.63 | 98.40 | 94.87 | 60.40 | 3.85 | **98.28** | 96.06 |
| | | MobileV2 | 85.03 | 98.00 | 10.42 | 55.69 | 4.07 | **91.00** | 50.88 |
| | | PARN101 | 88.30 | 98.80 | 5.47 | **66.11** | 10.16 | 48.39 | 27.67 |
| | | GoogLeNet | 92.94 | 98.20 | 7.48 | **74.75** | 3.57 | 66.60 | 37.89 |
| | WaNet | VGG-16 | 88.53 | 97.80 | 5.22 | 26.99 | 3.39 | 69.75 | **70.10** |
| | | RN-18 | 87.51 | 98.40 | **80.42** | 20.20 | 51.96 | 45.86 | 45.71 |
| | | MobileV2 | 85.75 | 96.40 | 6.50 | 14.37 | 4.28 | **86.29** | 45.04 |
| | | PARN101 | 88.46 | 98.40 | 4.09 | 15.02 | 3.38 | **28.14** | 12.28 |
| | | GoogLeNet | 92.70 | 99.80 | **96.16** | 49.20 | 11.09 | 94.90 | 64.21 |
| GTSRB | BadNets | VGG-16 | 97.58 | 100.00 | 5.53 | 11.64 | 9.90 | **99.99** | 99.98 |
| | | RN-18 | 98.12 | 100.00 | 19.44 | 8.55 | 9.36 | 41.69 | **99.98** |
| | | MobileV2 | 97.32 | 100.00 | 7.15 | 6.21 | 19.21 | **100.00** | 93.14 |
| | | PARN101 | 96.68 | 100.00 | 4.93 | 3.81 | 3.04 | **100.00** | 97.43 |
| | | GoogLeNet | 99.20 | 100.00 | 60.57 | 4.62 | 42.26 | 62.19 | **92.16** |
| ImageNet Subset | BadNets | RN-18 | 58.13 | 100.00 | **100.00** | 73.21 | 2.69 | **100.00** | 100.00 |
| | | EfficientNet-b0 | 66.41 | 100.00 | **100.00** | 32.35 | 2.70 | **100.00** | 100.00 |
| | ISSBA | RN-18 | 57.11 | 100.00 | 39.97 | 9.05 | 6.56 | 99.82 | 99.37 |
| | | EfficientNet-b0 | 62.19 | 100.00 | 33.57 | 14.18 | 3.97 | **99.88** | 98.80 |
| Average | - | - | 87.66 | 97.91 | 38.2 | 28.44 | 10.4 | **71.69** | 59.38 |

Table 10: The detection performance on test set measured by true positive rate (TPR) and false positive rate (FPR) for our CD method and the baselines against 12 backdoor attacks, reported as TPR/FPR. For detection methods, classification threshold $t$ is set to 1 standard deviation away from the mean calculated for 1% of the clean training set. The poisoning rate is 5% for all attacks. The best detection results are in **bold**, best is selected base on TPR first then use FPR for tie-breaking. TPR higher the better, FPR lower the better.

| Dataset | Attack | Model | AC | STRIP | SS | CD-L | CD-F |
|---|---|---|---|---|---|---|---|
| CIFAR10 | BadNets | VGG-16 | 99.60/68.14 | 87.60/11.18 | 19.40/68.04 | 100.00/10.97 | **100.00/7.93** |
| | | RN-18 | 99.80/51.98 | **100.00/8.95** | 93.80/79.08 | 100.00/14.54 | 100.00/14.33 |
| | | MobileV2 | 92.80/38.27 | 98.60/9.33 | 91.00/65.37 | **99.80/15.52** | 92.20/16.49 |
| | | PARN101 | 38.20/58.63 | **100.00/9.20** | 100.00/73.41 | 44.00/14.68 | 35.20/14.93 |
| | | GoogLeNet | 99.40/53.12 | **100.00/8.57** | 30.20/75.32 | 94.60/15.48 | 65.40/16.15 |
| | Blend | VGG-16 | **94.20/62.82** | 44.80/12.39 | 41.60/66.66 | 92.80/12.26 | 93.60/9.94 |
| | | RN-18 | 98.80/59.66 | 50.60/12.09 | **100.00/80.13** | 99.80/14.46 | 94.80/14.64 |
| | | MobileV2 | 20.00/35.73 | 66.00/11.71 | 97.00/64.02 | **97.40/16.32** | 69.00/16.76 |
| | | PARN101 | 51.40/52.78 | 77.00/10.59 | 46.60/73.94 | **97.40/14.71** | 91.80/15.17 |
| | | GoogLeNet | **93.80/50.84** | 71.80/11.66 | 46.40/67.93 | 37.60/15.53 | 81.00/16.05 |
| | CL | VGG-16 | 4.80/60.75 | 86.20/11.76 | 54.00/68.49 | 84.80/12.62 | **89.80/9.81** |
| | | RN-18 | 14.40/52.54 | **97.00/9.79** | 56.40/77.96 | 95.20/13.59 | 81.40/13.71 |
| | | MobileV2 | 16.00/36.38 | **97.00/9.97** | 67.00/66.81 | 72.60/16.82 | 46.80/17.07 |
| | | PARN101 | 5.80/54.40 | **96.40/9.53** | 64.20/72.77 | 52.20/13.36 | 47.60/13.99 |
| | | GoogLeNet | 0.00/43.00 | **88.40/10.48** | 65.20/70.57 | 28.40/14.89 | 22.80/14.75 |
| | DFST | VGG-16 | **99.40/66.74** | 10.80/14.66 | 0.60/55.87 | 82.60/11.73 | 85.80/10.75 |
| | | RN-18 | 97.20/58.31 | 8.40/13.60 | **100.00/80.61** | 95.80/14.20 | 91.20/13.96 |
| | | MobileV2 | **89.00/36.46** | 4.40/15.31 | 28.80/62.75 | 66.40/16.18 | 37.80/16.93 |
| | | PARN101 | **92.80/54.97** | 8.40/13.77 | 44.40/72.36 | 50.60/14.62 | 72.40/14.87 |
| | | GoogLeNet | 99.00/53.41 | 6.40/14.81 | **100.00/75.08** | 92.80/15.24 | 24.60/15.44 |
| | Dynamic | VGG-16 | 86.80/62.71 | 61.40/11.49 | 79.80/70.18 | **97.60/13.38** | 97.20/10.74 |
| | | RN-18 | 96.20/58.81 | 62.40/10.20 | 99.20/79.74 | **100.00/13.91** | 99.40/14.09 |
| | | MobileV2 | 78.40/38.21 | 91.80/7.65 | 84.60/65.57 | **97.40/15.51** | 84.60/16.23 |
| | | PARN101 | 60.00/53.23 | 83.20/9.20 | **89.60/73.20** | 80.60/13.61 | 66.20/14.71 |
| | | GoogLeNet | 99.00/53.68 | 94.80/7.74 | 59.60/71.93 | **100.00/15.66** | 96.80/15.82 |
| | FC | VGG-16 | **99.80/90.64** | 56.71/22.62 | 90.38/85.27 | 96.09/35.46 | 82.87/33.40 |
| | | RN-18 | 99.60/99.03 | 72.75/17.68 | 92.99/81.24 | **99.80/34.84** | 99.10/27.91 |
| | | MobileV2 | 96.79/90.31 | 66.23/16.70 | 55.11/81.99 | **99.70/33.07** | 98.30/30.31 |
| | | PARN101 | **99.90/93.91** | 71.94/17.45 | 95.49/74.02 | 99.70/33.34 | 98.60/29.83 |
| | | GoogLeNet | 99.70/99.05 | 53.51/19.48 | 97.90/73.22 | **100.00/36.57** | 99.10/31.09 |
| | Nashville | VGG-16 | 82.80/60.27 | 9.40/13.61 | 76.80/71.66 | 83.60/13.48 | **84.00/10.00** |
| | | RN-18 | 91.00/59.96 | 12.20/13.27 | **98.20/80.32** | 93.40/14.55 | 82.00/14.52 |
| | | MobileV2 | 97.60/35.34 | 8.60/13.96 | **99.60/65.47** | 99.40/15.45 | 98.60/16.27 |
| | | PARN101 | 67.40/50.68 | 9.40/14.06 | 96.00/74.09 | **96.60/14.55** | 88.80/14.69 |
| | | GoogLeNet | 98.40/44.71 | 11.80/14.85 | 100.00/71.71 | **100.00/16.05** | 89.40/15.75 |
| | SIG | VGG-16 | **69.60/63.18** | 23.20/13.35 | 43.20/63.29 | 39.00/12.20 | 40.00/8.61 |
| | | RN-18 | 98.20/53.76 | 37.00/12.46 | **100.00/80.44** | 99.80/12.45 | 99.80/13.72 |
| | | MobileV2 | 14.20/37.92 | 12.00/14.71 | 71.60/67.08 | 83.20/15.64 | **91.40/16.76** |
| | | PARN101 | 93.40/61.20 | 32.60/12.03 | 90.40/62.49 | 93.40/12.99 | **98.60/14.17** |
| | | GoogLeNet | 51.00/43.82 | 15.80/13.74 | 88.20/72.17 | 88.20/14.95 | **96.40/15.61** |
| | Smooth | VGG-16 | **83.20/54.71** | 36.80/12.36 | 34.40/67.78 | 63.60/13.22 | 28.00/7.89 |
| | | RN-18 | 91.00/52.63 | 41.80/11.60 | **100.00/80.20** | 88.00/14.54 | 67.60/14.58 |
| | | MobileV2 | 78.40/38.07 | 35.00/13.16 | 43.60/70.06 | **86.20/16.12** | 60.60/17.11 |
| | | PARN101 | **80.80/58.97** | 18.80/12.71 | 51.20/70.69 | 61.00/13.99 | 40.40/14.54 |
| | | GoogLeNet | 90.80/51.97 | 10.80/14.67 | **99.60/73.55** | 88.00/15.51 | 77.40/15.85 |
| | Trojan | VGG-16 | 94.80/62.41 | 75.80/10.99 | 70.60/69.80 | **97.00/12.78** | 96.80/8.48 |
| | | RN-18 | 98.00/61.13 | 83.60/9.80 | **99.60/79.44** | 98.40/13.66 | 97.40/14.40 |
| | | MobileV2 | 71.60/36.98 | 78.20/10.20 | 57.20/64.95 | **96.80/15.35** | 82.60/16.67 |
| | | PARN101 | 88.20/57.54 | 87.00/9.28 | **95.40/74.43** | 84.60/14.45 | 68.60/15.14 |
| | | GoogLeNet | 91.20/55.22 | 86.40/8.99 | 50.60/70.56 | **92.40/15.02** | 72.40/15.64 |
| | WaNet | VGG-16 | 81.60/58.32 | 68.00/11.17 | 81.00/70.65 | **92.80/12.17** | 81.60/11.13 |
| | | RN-18 | 93.60/56.32 | 55.00/11.93 | **100.00/80.69** | 99.40/10.73 | 76.60/13.82 |
| | | MobileV2 | 54.40/36.92 | 43.00/11.80 | 92.80/68.62 | **98.20/15.09** | 78.00/16.17 |
| | | PARN101 | 51.80/53.60 | 46.00/12.06 | 69.60/74.51 | **78.60/15.13** | 44.80/13.51 |
| | | GoogLeNet | 98.40/52.62 | 87.40/10.51 | **100.00/74.91** | 99.80/14.82 | 93.00/16.26 |
| GTSRB | BadNets | VGG-16 | 95.88/65.53 | 33.44/10.15 | 100.00/56.86 | **100.00/4.88** | 86.21/0.00 |
| | | RN-18 | 100.00/92.02 | 29.16/12.63 | 100.00/68.26 | 100.00/11.06 | **100.00/0.54** |
| | | MobileV2 | 99.21/72.83 | 25.36/18.39 | 100.00/61.14 | **100.00/2.06** | 79.56/0.00 |
| | | PARN101 | 98.57/78.88 | 2.22/15.74 | 90.17/75.54 | **100.00/7.85** | 98.89/2.90 |
| | | GoogLeNet | 100.00/81.97 | 14.10/16.13 | 100.00/79.63 | **100.00/9.68** | 27.26/0.06 |
| ImageNet Subset | BadNets | RN-18 | 100.00/53.11 | 99.80/13.35 | 61.00/71.81 | 100.00/32.99 | **100.00/22.15** |
| | | EfficientNet-b0 | **100.00/10.61** | 90.00/15.49 | 0.40/70.23 | 100.00/20.07 | 100.00/20.27 |
| | ISSBA | RN-18 | 87.60/48.92 | 38.40/17.84 | 77.20/77.42 | 100.00/53.62 | **100.00/33.12** |
| | | EfficientNet-b0 | 75.00/28.48 | 56.00/21.14 | 64.40/70.81 | 100.00/40.25 | **100.00/27.25** |
| Average | - | - | 79.54/56.68 | 65.71/27.38 | 75.00/71.98 | **88.39/16.92** | 79.25/15.26 |

# D    ADDITIONAL VISUALIZATIONS

Figures 18 to 31 show the randomly selected input images, learned mask, and distilled CP of both clean and backdoored samples for each corresponding experiment evaluated in section 4. The masks and CP are extracted from the corresponding backdoored model (ResNet-18) using CD-L.

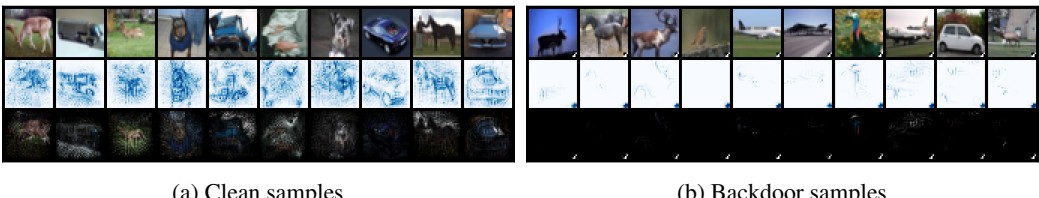

| (a) Clean samples | (b) Backdoor samples |

Figure 18: Corresponding input images, learned masks and CPs for BadNets (Gu et al., 2017) on CIFAR10.

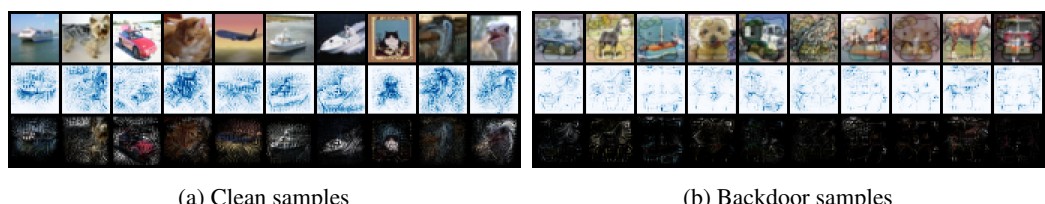

| (a) Clean samples | (b) Backdoor samples |

Figure 19: Corresponding input images, learned masks and CPs for Blend (Chen et al., 2017) on CIFAR10.

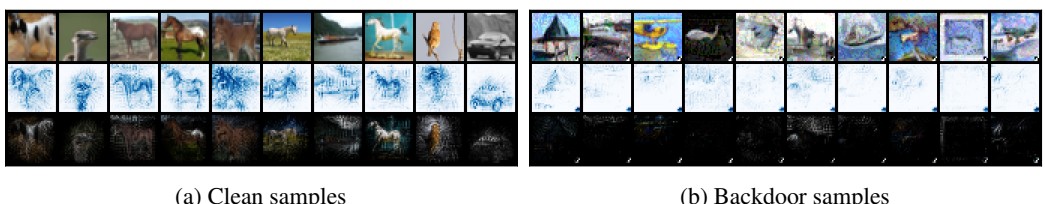

| (a) Clean samples | (b) Backdoor samples |

Figure 20: Corresponding input images, learned masks and CPs for CL (Turner et al., 2018) on CIFAR10.

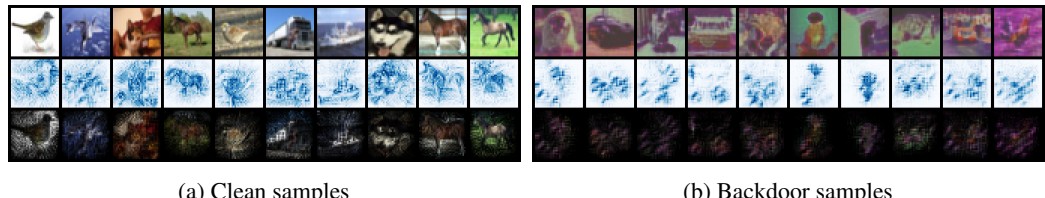

| (a) Clean samples | (b) Backdoor samples |

Figure 21: Corresponding input images, learned masks and CPs for DFST (Cheng et al., 2020) on CIFAR10.

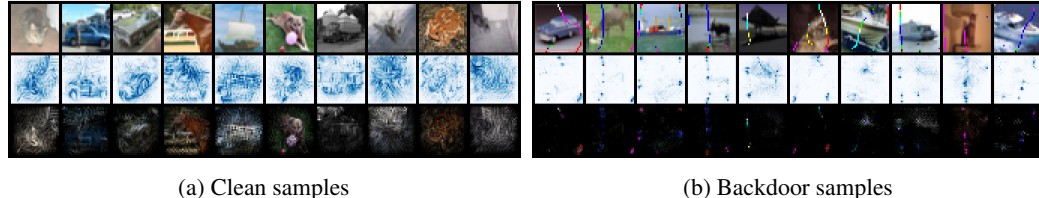

(a) Clean samples

(b) Backdoor samples

Figure 22: Corresponding input images, learned masks and CPs for Dynamic (Nguyen & Tran, 2020) on CIFAR10.

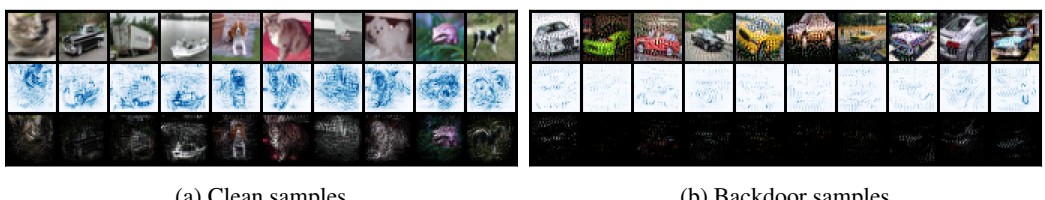

(a) Clean samples

(b) Backdoor samples

Figure 23: Corresponding input images, learned masks and CPs for FC (Shafahi et al., 2018) on CIFAR10.

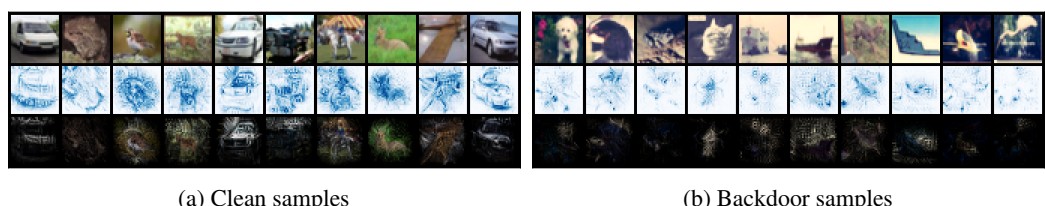

(a) Clean samples

(b) Backdoor samples

Figure 24: Corresponding input images, learned masks and CPs for Nashville (Liu et al., 2019) on CIFAR10.

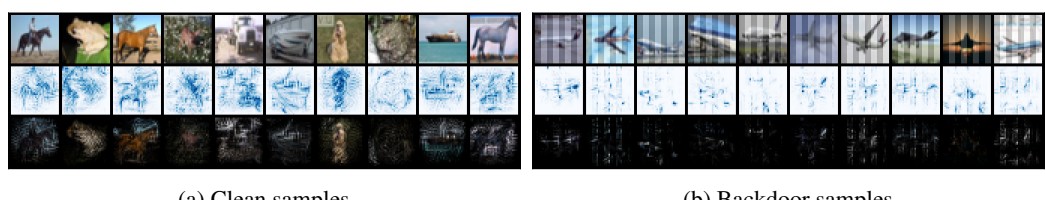

(a) Clean samples

(b) Backdoor samples

Figure 25: Corresponding input images, learned masks and CPs for SIG (Barni et al., 2019) on CIFAR10.

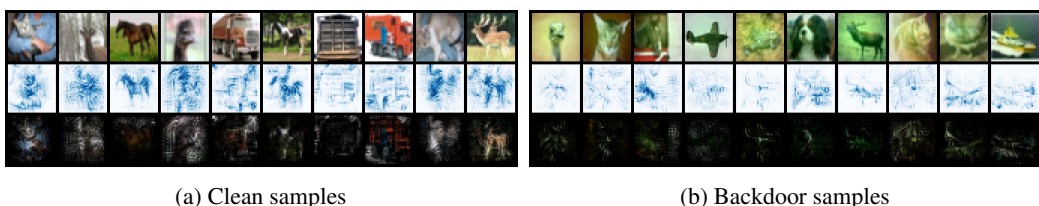

(a) Clean samples

(b) Backdoor samples

Figure 26: Corresponding input images, learned masks and CPs for Smooth (Zeng et al., 2021) on CIFAR10.

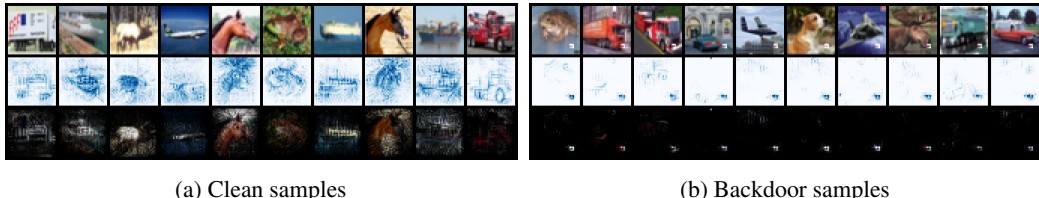

(a) Clean samples                              (b) Backdoor samples

Figure 27: Corresponding input images, learned masks and CPs for Trojan (Liu et al., 2018b) on CIFAR10.

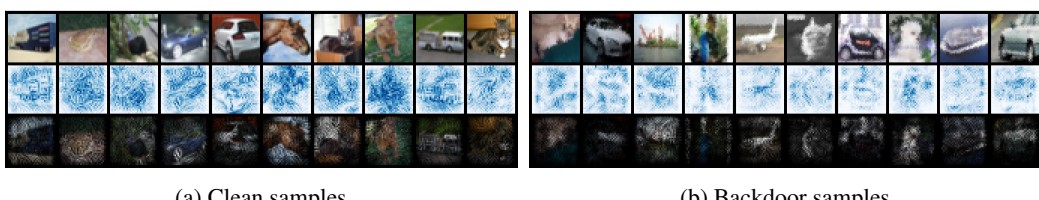

(a) Clean samples                              (b) Backdoor samples

Figure 28: Corresponding input images, learned masks and CPs for using WaNet trigger (Nguyen & Tran, 2021) on CIFAR10.

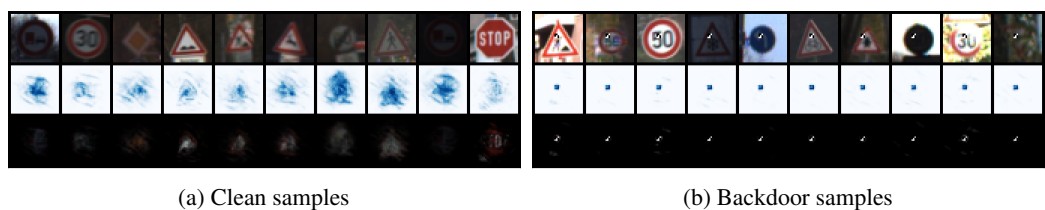

(a) Clean samples                              (b) Backdoor samples

Figure 29: Corresponding input images, learned masks and CPs for BadNets (Gu et al., 2017) on GTSRB.

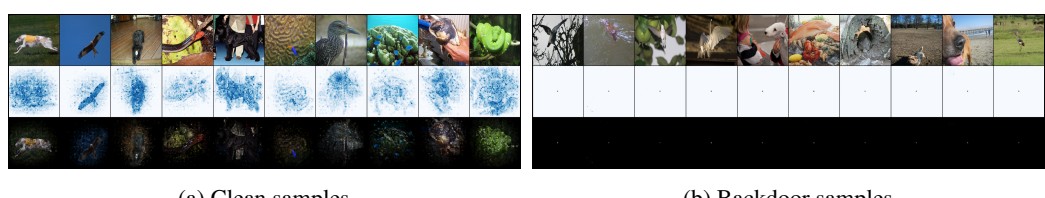

(a) Clean samples                              (b) Backdoor samples

Figure 30: Corresponding input images, learned masks and CPs for BadNets (Gu et al., 2017) on ImageNet subset.

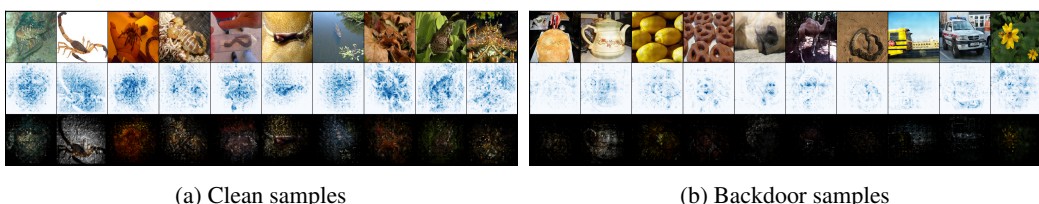

(a) Clean samples                              (b) Backdoor samples

Figure 31: Corresponding input images, learned masks and CPs for ISSBA (Li et al., 2021c) on ImageNet subset.

