# OpenReview forum: "Distilling Cognitive Backdoor Patterns within an Image"
_ICLR.cc/2023/Conference — ICLR 2023 poster_

### Official Review · Reviewer_Xw5r · 2022-10-14

**Confidence:** 3
**Correctness:** 4
**Technical Novelty And Significance:** 3
**Empirical Novelty And Significance:** 3
**Recommendation:** 8

**Clarity, Quality, Novelty And Reproducibility:**

As mentioned in the Strengths, the work is well organized, clearly states related work and contributions. This helps understand the novetly which seems reasonable. Based on the details in Section 3 and experimental setup in 4, I'm reasonably confident this is reproducible.

**Details Of Ethics Concerns:**

None that come to mind

**Strength And Weaknesses:**

Strengths
* The paper is clearly written, well organized and easy to follow.
* The related work is well motivated.
* The results and ablation studies directly answer the authors claims

Weakness
* Nit: The theoretical motivation could be expanded in Section 3.1. How were each of the terms generalized and weighted?
* The case study on based on relatively simple datasets, it would be great to see this expanded on diverse distributions.
* It would be interesting to see this approach not only in image space, but including temporal dependencies such as in videos.

**Summary Of The Paper:**

The given work proposes their "Cognitive Distillation" (CD) approach for distilling, detecting backdoor patterns from pixels. The author's claim that this is able to extract the minimal pattern responsible for a model's prediction. Based on their studies, they argue that a sparse input pattern is responsible for backdoor predictions regardless of the trigger pattern. They evaluate their approach on various architectures, standard datasets and attack types. The authors argue that could help detect potential bias in real world datasets.

**Summary Of The Review:**

To the best of my knowledge, the authors provide a reasonable contribution to mitigate adversarial backdoor patterns. The work is well written, with the empirical evaluation answering the questions asked. The work seems reasonably novel clearly stating the related work, would be interesting to see it expanded to challenging benchmark tasks. Overall, recommending an accept.

---

> ### Author Response · Authors · 2022-11-10
> **Response to reviewer Xw5r**
>
> Thanks for reviewing our work and the valuable comments. We hope the following responses could help address your concerns.
>
> ---
> **Q1:** How were each of the terms (in Eq. (1) and (2)) generalized and weighted?
>
> **A1:** Please find an ablation study of each regularization term in Appendix B.6 and B.7. $\beta$ and $TV$ could help the pattern to align with human perception, which only affects the detection performance slightly.  $\alpha$ and $|m|_1$ are for finding the minimal sufficient patterns and are key for accurate detection. It empirically shows that our CD can effectively detect backdoor samples within a reasonable range of $\alpha$, e.g., [0.05, 0.005].
>
> ---
>
> **Q2:** Other datasets for bais detection
>
> **A2:** Thanks for the suggestion. We chose CelebA to conduct our case study because it contains several well-known biases previously established in the literature.  It thus provides the ground truth to verify the potential biases (strong correlations between certain attributes to a target class) detected by our CD. In fact, we plan to expand this case study into a large-scale bias investigation over a wide range of datasets and models. We will leave this to our future work.
>
> ---
>
> **Q3:** Experiments on videos
>
> **A3:** Thanks for the suggestion. We will explore more types of data in our future work while focusing on images in this work. We agree that the regularization terms of CD should be adapted to model different types of features.

---

> ### Author Response · Authors · 2022-11-23
> **Thanks for your feedback**
>
> Dear reviewer Xw5r,
>
> Thanks very much for your recognition of our work and the encouraging comments.
> They have greatly improved our work. Please feel free to let us know if you have more questions.

---

### Official Review · Reviewer_o4Nd · 2022-10-25

**Confidence:** 3
**Correctness:** 3
**Technical Novelty And Significance:** 3
**Empirical Novelty And Significance:** 3
**Recommendation:** 6

**Clarity, Quality, Novelty And Reproducibility:**

The writing is clear. The proposed idea is new and interesting. However, I have some concerns about the evaluation metrics, the mitigated clean accuracy, and some defense results, as mentioned in the Weaknesses part.

The results seem to be reproducible since the code is submitted in the supplementary.

**Strength And Weaknesses:**

### Strengths
- The paper proposes a new idea of extracting the minimal essential pattern for each input image. It points out that such a pattern is often exceptionally small when the model and the input data are poisoned.
- Based on that observation, the authors proposed a new method to detect backdoor samples. Such a detection scheme provides high AUC scores when detecting backdoor samples generated by various backdoor attacks on CIFAR-10.
- The paper then proposes to mitigate the model's backdoor effects via Anti-backdoor learning, which effectively reduces the models' ASRs.
- CD is also used to detect potential biases on a face attribute classification dataset.

### Weaknesses
- FAR should be normalized by the poisoning rates. For example, given a dataset with 5% data poisoned, a random guess would provide a FAR of 5%, which sounds low but actually not.
- Given the data imbalance (poisoned vs. clean) and the issue mentioned above, PR curves are preferable over ROC curves, and AUC-PRs are preferable over AUC-ROCs. While TRR and FAR are helpful metrics, Recall is the most critical metric (the percentage of poisoned data that can be detected and removed).
- The mitigated clean accuracy values in Fig.4 are too low, given the original numbers are often around 88-92% (Table 7). Given that significant accuracy drop, these mitigation approaches are impractical.
- Also, the authors should conduct experiments on clean datasets to see how many clean examples are falsely detected and how much it affects the mitigated clean accuracy.
- The paper provides a massive amount of experiments. However, most experiments are conducted on CIFAR-10.
- In their original papers, recent backdoor attacks such as Dynamic and WaNet can easily pass STRIP by providing similar entropy histograms as from the clean model counterparts. However, this paper's detection AUCs for STRIP on these attacks are unexpectedly high. Can the authors explain the reason?
- The defense baselines are pretty old. The authors should include some more recent backdoor detection approaches, such as [1].
- While interesting, the bias detection part is irrelevant to the rest of the paper.

[1]. Zeng Y, Park W, Mao ZM, Jia R. Rethinking the backdoor attacks' triggers: A frequency perspective. In ICCV 2021 (pp. 16473-16481).

**Summary Of The Paper:**

The paper proposes a novel backdoor defense called Cognitive Distillation (CD) that defends against data-poisoning attacks. Given a suspect model, CD extracts the minimal essential pattern, called Cognitive Pattern (CP), responsible for the model's prediction on each image input. It observes that backdoored models often provide exceptionally small CPs on poisoned inputs, and this can be used to detect backdoor examples. Then, CD can employ Anti-backdoor learning on the purified dataset to mitigate the model's backdoor effects. CD provides high AUC scores when detecting backdoor samples generated by various backdoor attacks on CIFAR-10. In backdoor mitigation, CD reduces the ASRs better compared with the previous mitigation baselines. CD is also used to detect potential biases on a face attribute classification dataset.

**Summary Of The Review:**

The proposed idea is new and interesting. However, I have some concerns about the evaluation metrics, the mitigated clean accuracy, and some defense results, as mentioned in the Weaknesses part.

---

> ### Author Response · Authors · 2022-11-10
> **Response to reviewer o4Nd**
>
> Thanks for your thoughtful comments. Please kindly find our response below to your concerns.
>
> ---
> **Q1:** FAR should be normalized by the poisoning rates
>
> **A1:** Thanks for the suggestion. We have added the suggested normalized FAR as $|x_{bd} \in \mathcal{D}_c’|/|\mathcal{D}_b|$ in Figure 4 where it shows a more noticeable improvement over the baselines.
>
> ---
> **Q2:**  AUC-PRs are preferable over AUC-ROCs
>
> **A2:** Thanks for the suggestion. We have added the area under the precision-recall curve (AU-PRC) in appendix C (Table 7 and Table 9 for the detailed results on different models, attacks, and datasets) and summarized the result on CIFAR-10 (for training/test set, respectively) in the table below. As can be observed, our CD outperforms the baselines by an even larger margin as compared to AU-ROC.
>
> |  Attacks  |     ABL     |      AC     |  Frequency  |    STRIP    |      SS     |       CD-L      |     CD-F    |
> |:---------:|:-----------:|:-----------:|:-----------:|:-----------:|:-----------:|:---------------:|:-----------:|
> |  BadNets  |   26.13/-   | 38.99/22.50 | 36.67/32.47 | 60.81/57.70 | 38.94/18.01 | **75.53/75.48** | 60.09/60.18 |
> |   Blend   |   42.82/-   | 37.93/36.80 | 13.02/12.22 | 26.55/24.15 |  15.54/5.11 | **68.99/67.97** | 44.50/44.97 |
> |     CL    |   38.93/-   |  20.61/3.18 | 83.42/32.47 | 72.51/46.84 |  3.91/4.00  | **88.77/46.90** | 66.86/35.77 |
> |    DFST   | **49.19**/- | 48.12/46.86 | 24.57/23.21 |  5.40/5.48  |  25.85/6.48 | 47.34/**48.47** | 32.25/32.46 |
> |  Dynamic  |   46.65/-   | 48.15/45.53 | 75.03/74.76 | 64.35/59.29 | 21.92/10.72 | **88.89/87.65** | 70.95/69.09 |
> |     FC    |   39.88/-   | 51.80/79.71 | 82.50/87.78 | 38.57/39.61 | 38.09/14.78 | **89.48/90.34** | 72.74/81.15 |
> | Nashville |   26.24/-   | 63.88/39.04 |  11.76/9.54 |  5.70/4.59  | 30.57/10.57 | **89.44/87.60** | 66.46/65.02 |
> |    SIG    | **79.02**/- | 67.07/22.18 |  8.92/6.05  |  22.31/7.14 | 25.56/10.59 | 71.93/55.14 | 71.11/**62.72** |
> |   Smooth  |   27.41/-   | 53.83/40.09 |  5.90/5.10  |  10.49/8.72 | 41.67/12.75 | **55.66/52.26** | 24.34/23.32 |
> |   Trojan  |   26.30/-   | 25.07/24.78 | 63.81/62.89 | 59.38/59.35 |  15.58/4.96 | **80.56/80.12** | 60.54/61.85 |
> |   WaNet   |    5.69/-   | 37.77/38.48 | 61.22/61.01 | 28.01/25.16 | 30.78/14.82 | **63.58/64.99** | 44.51/47.47 |
> |    Avg    |   37.12/-   | 44.84/36.29 | 42.44/37.05 | 35.82/30.73 | 26.22/10.25 | **74.56/68.81** | 55.85/53.09 |
>
>
> ---
>
> **Q3:** The mitigated clean accuracy values in Fig.4 are too low.
>
> **A3:** Thanks for pointing this out. In our initial submission, we only applied the key ABL mitigation technique while omitted the additional finetuning step (a 60 epochs of finetuning on the predicted clean subset) [2]. We have now added the additional finetuning step back and updated Figure 4 accordingly. The clean accuracy is now on par with the original accuracy. Note that this does not affect TRR/FAR but only the clean accuracy and ASR after the mitigation.
>
> ---
> **Q4:** The authors should conduct experiments on clean datasets to see how many clean examples are falsely detected and how much it affects the mitigated clean accuracy.
>
> **A4:** Thanks for the insightful suggestion. We have trained a model (ResNet-18) on clean CIFAR-10, then applied different detection methods to detect and remove (potential) backdoor examples for removal, and retrained a new model on the purified dataset. This simulates the scenario where the defender collects a clean dataset from an untrusted source but suspects that it may contain backdoor samples. In this case, clean accuracy is arguably the primary concern. We remove 10% to 40% "potential backdoor samples" from the original dataset. The experiments are repeated for 3 times with different random seeds, and the mean and standard deviations results are reported in the table below. “Random” denotes randomly removing samples from the dataset. As can be observed, the clean accuracy is largely preserved if less than 20% of the samples are removed. While this result is rather dataset-dependent, note that 20% or higher is an extremely high poisoning rate. A practical recommendation is to remove a number of samples that do not cause a significant drop in clean accuracy.
>
>
> |  Method  |      10%       |      20%       |      30%       |     40%       |
> |----------|----------------|----------------|----------------|----------------|
> |   ABL    | 87.86 ± 0.30 | 87.58 ± 0.40 | 86.66 ± 0.41 | 85.62 ± 0.36 |
> |    AC    | 87.50 ± 0.40 | 86.56 ± 0.33 | 85.65 ± 0.59 | 84.17 ± 0.28 |
> |  STRIP   | 87.82 ± 0.22 | 86.49 ± 0.52 | 85.12 ± 0.23 | 83.11 ± 0.42 |
> |    SS    | 87.65 ± 0.09 | 86.71 ± 0.11 | 85.60 ± 0.35 | 84.47 ± 0.33 |
> |  Random  | 87.37 ± 0.28 | 86.09 ± 0.37 | 84.76 ± 0.26 | 83.55 ± 0.17 |
> |   CD-L   | 87.30 ± 0.18 | 87.06 ± 0.55 | 85.29 ± 0.40 | 82.62 ± 0.89 |
> |   CD-F   | 87.51 ± 0.22 | 86.92 ± 0.26 | 85.48 ± 0.31 | 83.72 ± 0.04 |

---

> ### Author Response · Authors · 2022-11-10
> **Response to reviewer o4Nd**
>
> ---
> **Q5:** Most experiments are conducted on CIFAR-10.
>
> **A5:** We understand the concern. However, previous backdoor attacks have released their trigger patterns or generators for the CIFAR-10 dataset. For fair comparisons and also with a view to a comprehensive understanding on the most representative dataset, we reported the results on CIFAR-10 as our main result. However, we also conducted experiments on GTSRB and a 200-class ImageNet subset. Please note that many attacks are not directly transferable to ImageNet or the ASR cannot be guaranteed. Our results demonstrate that our CD is also effective on other datasets (ImageNet subset and GSTRB) and performs even better on higher-resolution images. Figure 2 (I) shows the distribution of the L1 norm on ISSBA (ImageNet with resolution $224 \times 224$). Table 1 shows that our detection can successfully detect backdoor samples with higher resolutions. The visualizations can be found in Figures 29 and 30 in Appendix D. With higher resolutions, the separation in L1 norm of the masks is more distinct. We are happy to include more experiments if the reviewer believes there are more attacks which should be tested on ImageNet.
>
> ---
> **Q6:** Dynamic and WaNet bypass STRIP.
>
> **A6:** In the original papers of Dynamic and WaNet, they manipulated the training objective to bypass STRIP. However, in this work, we mainly focus on the data poisoning threat model and assume that the adversary cannot manipulate the training procedure. Our results indicate that poisoning with Dynamic and WaNet triggers alone can effectively inject the backdoor, but this may not be enough to evade the STRIP detection without modifying the training loss.
>
> ---
> **Q7:** The authors should include some more recent backdoor detection approaches, such as [1].
>
> **A7:** Thanks for the suggestion. We have now added the frequency detection method in [1] as one of our baselines (see Table 1 and Figure 3/4). More details can be found in Appendix B.4. The average detection AU-ROC of [1] over different attacks and models is 84.62/82.51 on the training/test set, respectively.  Note the corresponding performance of our CD-L is 96.44/94.79, which is ~10% higher.
>
>
> ---
> **Q8:** The bias detection part is irrelevant to the rest of the paper.
>
> **A8:** The main focus of the paper is backdoor sample detection and that we included the bias detection as an application which can now be linked to backdoor sample detection through the use of our CD technique.  In other words, use of CD reveals the similarity between the two fields of backdoor sample detection and bias detection.
> Inspired by the similarity between bias and backdoor, that is, they both trigger over-easy predictions based on only a tiny amount of information perceived from the input. And our CD has the potential to be an effective and efficient tool for automated bias detection just like its demonstrated strong capability in backdoor sample detection. We beleive this is extremely beneficial to the other important topic in trustworthy machine learning: fair/debiased learning. We have clarified this point in the updated paper.
>
> ---
> [1] Zeng Y, Park W, Mao ZM, Jia R. Rethinking the backdoor attacks' triggers: A frequency perspective. In ICCV 2021.
>
> [2] Li, Y., Lyu, X., Koren, N., Lyu, L., Li, B., & Ma, X. Anti-backdoor learning: Training clean models on poisoned data. NeurIPS 2021

---

> ### Comment · Reviewer_o4Nd · 2022-11-16
> **I have increased my score**
>
> I am satisfied with most of the authors' answers. I still have some doubts on A6, and the algorithm is designed based on an empirical observation. Hence, I decided to set my score as 6.

---

> ### Author Response · Authors · 2022-11-23
> **Thanks for your feedback**
>
> Dear reviewer o4Nd,
>
> Thank you very much for your encouraging feedback. Your quality and timely review is much appreciated. For Q6, we will conduct a thorough investigation and update the results in the next revision.

---

### Official Review · Reviewer_czZL · 2022-10-25

**Confidence:** 4
**Clarity, Quality, Novelty And Reproducibility:** I included this evaluation in my comm…
**Correctness:** 1
**Technical Novelty And Significance:** 2
**Empirical Novelty And Significance:** Not applicable
**Recommendation:** 3

**Details Of Ethics Concerns:**

No concern.

**Strength And Weaknesses:**

Strengths:

1. The paper empirically analyzes the commonality across the backdoor attacks.
2. The paper exploits the commonality to build a backdoor input detection method.
3. The paper discusses some other useful ways to harness CD.


Weaknesses:

1. It's not convincing (from the paper) that the backdoored model only uses a subset of the trigger.
2. The technical novelty of the proposed method (CD) is weak (CD is an adaptation of Neural Cleanse).
3. The evaluation is weak (varying poisoning rates won't reflect adaptive attacks, and Neural Cleanse may perform better under the same setting).


Detailed comments:

The paper studies a way to capture input samples containing potential triggers that may lead to backdoor behaviors. As backdoor attacks are a growing concern, it is important to study defense mechanisms to reduce the attack surface. This paper strives to find the commonality across the backdoor behaviors.

However, the major problem of this approach is that the defense relies on an assumption (backdoored models will use a subset of trigger patterns for classification) that can be easily broken. Another concern is that the optimization process (which is the paper's technical contribution) is highly similar to the process proposed by Wang et al., Neural Cleanse. The paper (moreover) somewhat lacks the evaluation against adaptive attacks, and the discussion about bias detection comes out of the blue. Thus, I think this paper is not ready for ICLR 2023.

My detailed comments can be found below:


[On the Common Observation]

The paper's claim depends on the observation that most backdoored models rely on a small subset of pixels over the entire trigger pattern. I don't think this is true, and the paper studies only a subset of backdoor attacks. I assume that this may come from the way we train our models. However, as the attacker controls the entire training process, they can add an objective that hinders backdoor behaviors in the (partial) presence of the trigger pattern the attacker uses. By doing so, this defense cannot detect any backdoor inputs at all.

Moreover, even if we just assume that the backdoor attacks "only" use a small subset of pixels in the trigger pattern. There is a problem that the proposed CD can "exactly" reconstruct the pattern used by the attacker. Prior work [1]---that proposed a similar reconstruction method---showed some negative results regarding this. A backdoored model can contain multiple trigger patterns that aren't injected by the attacker. If CD reconstructs one of those patterns, can we say that CD detects backdoored input?

In the long run, we may forecast a diminishing return. Are we sure whether a clean classifier does not naturally learn such small patterns? As the resolution increases (224+), I assume that there are several minor details in the dataset that a model can learn and focus on. In this case, can we say that a model is backdoored, or does it learn the correct features? Just relying on the assumption that a mask will be small won't lead to a successful defense in the end.

[1] Sun et al., Poisoned classifiers are not only backdoored, they are fundamentally broken.


[On the Technical Novelty]

The optimization process proposed by this work has already been exploited in the backdoor literature. Neural Cleanse proposed a trigger reconstruction process that relies on the same assumption; the pattern will be composed of a smaller number of pixels. [1] also proposed a reconstruction process (that seems to be more advanced than this paper uses, as they use smoothing with a denoiser). Thus, the novelty is less strong.


[On the Adaptive Attacks and Bias Detection]

As pointed out above, the attacker can train a model with an objective that forces the model to use all the pixels in the trigger pattern to trigger backdoor behaviors. By combining this objective and a pattern that uses all the input pixels (such as the trojan trigger used in the literature), the attacker could evade this defense completely.

It's nice that this paper discusses other use cases of CD. However, it's a bit unclear what scientific advances CD makes over the prior approaches to detecting biases in the training data. Unfortunately, in the evaluation, the paper does not draw any comparison with the baseline bias detection. Thus, I can make a conjecture that CD can detect biases, but I am unsure whether CD will have a novelty over the prior work.


[Minors]

1. (In the intro) References to the work that formulates backdooring as a multi-task learning is missing. It has been studied and presented in the "Blind Backdoors in Deep Learning Models" paper.

**Summary Of The Paper:**

This paper proposes a defensive mechanism, Cognitive Distillation (CD), that extracts potential trigger patterns (and a mask) responsible for the model's prediction. The paper starts with the observation that, even if the trigger pattern consists of multiple pixels and/or is complex, a few pixel values correspond to activating backdoors. The paper exploits this observation and proposes an optimization that reconstructs a potential trigger pattern and a mask from an input image. Using those reconstruction outputs, the paper identifies whether the input contains the backdoor or not (if the mask is sufficiently small, then it contains a backdoor). In evaluation, the paper compares the detection rate with four baseline defenses. The paper also discusses the possibility of using CD for bias detection.

**Summary Of The Review:**

This paper studies a method to capture input samples containing potential triggers that may lead to backdoor behaviors. The proposed method relies on the empirical observation that backdoor models only use a subset of pixels in the actual trigger pattern. However, the major problems of this approach are as follows:

(1) The assumption can be easily broken.
(2) The optimization process (which is the paper's technical contribution) is not novel.
(3) The paper lacks the evaluation against adaptive attacks
(4) It's unclear about the novelty of bias detection.

Thus, I believe the weaknesses outweigh the strengths of this paper.

---

> ### Author Response · Authors · 2022-11-10
> **Response to reviewer czZL**
>
> Thanks for reviewing our paper and the detailed comments. Please find our responses below.
>
> ---
> **Q1:** It's not convincing (from the paper) that the backdoored model only uses a subset of the trigger. The assumption can be easily broken.
>
> **A1:** There seems to be a misunderstanding. What we found is that only a small part of the trigger (e.g., a few pixels of a large background trigger pattern) is responsible sufficient for most of the backdoor prediction. This is more like an empirical finding based on the experiments with 12 backdoor attacks, rather than an assumption. We are happy to verify this finding on more attacks if the reviewer has further suggestions.
>
> **Different results as in [1]:**
>
> We would like to clarify that the work [1] mentioned by the reviewer studied a different threat model from ours. [1] exploited robust randomized training to finetune a backdoored model, in order to reconstruct an **alternative** trigger to control the backdoored classifier. Their objective is very different to ours. In particular, the objective of [1] is based on an adversarial attack under the  $L_2$ constraint (as defined in Eq. 1, 2, and 4 in [1]) to perturb samples into a target class, while our objective does not modify the prediction outcome (defined in Eq. 1 and 2).
>
> Also different from [1], which was only tested on patch-based attacks, our CD has been evaluated on a more diverse set of trigger patterns crafted by 12 different attacks (summarized in Table 4), and it can detect these attacks with high AUC. For example, our CD can successfully detect WaNet (denoted as image warping in [1]), and Blend attacks (denoted as image blending in [1]), yet both attacks failed in [1].
>
>
> **Higher resolutions:**
>
> We agree that it is possible that natural high-resolution images may contain small predictive patterns but they are not backdoor triggers. However, we believe that if such patterns exist, they should be carefully reviewed, as the model may not be making decisions based on causation/logic corresponding to human intuition. Taking a closer look at the visualizations in Figures 29/30, we see that the clean model generally perceives significantly large cognitive patterns (CPs) from the high-resolution ImageNet images ($224\times224$). This suggests that small CPs may indeed be unusual on high-resolution images and therefore should be treated as potential backdoors or biases.
>
> This is also why we applied our CD method to identify biases, since both backdoors and biases cause a shortcut correlation between a small CP and an overconfident prediction. In fact, a key point of our paper is to highlight this shared intrinsic property of shortcut predictions, and draw the community’s attention to the exceptionally small CPs previously hidden, but now uncoverable by our CD technique.
>
> ---
> [1] Mingjie Sun, Siddhant Agarwal, J. Zico Kolter., 2020. Poisoned classifiers are not only backdoored, they are fundamentally broken. arXiv preprint arXiv:2010.09080.

---

> ### Author Response · Authors · 2022-11-10
> **Response to reviewer czZL**
>
> ---
> **Q2:** The technical novelty of the proposed method (CD) is weak (CD is an adaptation of Neural Cleanse).
>
> **A2:** Please allow us to clarify that our CD and NC are solving two different problems. Their technical differences are as follows:
>
> - NC is a method that detects **backdoor models** while our CD aims to detect **backdoor samples**. These are two different tasks.
> - NC finds a pair of pattern and mask that can perturb all samples from other classes into the target class, like a universal adversarial pattern [2] for the target class. Our CD finds a mask for an image that highlights the most essential pixels responsible for the model’s prediction. There is no “attack” in our CD approach.
> - NC takes a pre-trained model and a set of training images (and their labels), and outputs $K$ pairs of masks and trigger patterns, where K is the number of classes. However, backdoor sample detection requires a score for each sample, rather than $K$ pairs of masks and trigger patterns. NC can compute the score for each class (for backdoor label detection) but not for the sample.
> Our CD takes a pre-trained model and an image, and returns a corresponding mask without using any labels. This means that our CD can be applied at test time where there are no labels. Our CD is also very efficient as its run time does not scale with the number of classes, but NC does.
> - NC and CD have completely different optimization objectives (NC uses a cross-entropy loss defined w.r.t. a target label, while our CD uses the absolute difference w.r.t. original prediction). In terms of the mask, NC replaces the (1-m) area with clean samples from different classes, while our CD replaces the (1-m) area with $\delta$ that is randomly sampled at each optimization step.
> NC does not work for attacks that use sample-specific triggers, which is not the case for our CD (as shown in Table 1 and Figure 3) which is effective.
>
> To summarize, our method differs from NC in terms of the objective, the optimization algorithm, as well as the outcome. A more detailed comparison between our CD and existing works (including NC, LIME and MP) can be found in Appendix A. Please also note that, as NC cannot detect backdoor examples, we did not consider NC as a baseline. If the reviewer has concrete suggestions of how to adapt NC for backdoor example detection, we are more than happy to include it in our main results.
>
> We have also made a Figure to differentiate the two techniques, which can be found anonymously at [https://anonymous.4open.science/r/iclr-cd-403C/NC%20V.S.%20CD.png](https://anonymous.4open.science/r/iclr-cd-403C/NC%20V.S.%20CD.png).
>
> ---
> [2] Moosavi-Dezfooli, Seyed-Mohsen, Alhussein Fawzi, Omar Fawzi, and Pascal Frossard. Universal adversarial perturbations. In CVPR, 2017.

---

> ### Author Response · Authors · 2022-11-10
> **Response to reviewer czZL**
>
> ---
> **Q3:** The evaluation is weak (varying poisoning rates won't reflect adaptive attacks, and Neural Cleanse may perform better under the same setting). The attacker can train a model with an objective that forces the model to use all the pixels in the trigger pattern to trigger backdoor behaviors. By combining this objective and a pattern that uses all the input pixels (such as the trojan trigger used in the literature), the attacker could evade this defense completely.
>
> **A3:** We believe the reviewer has misunderstood our claims. We did NOT claim that varying poisoning rates can reflect adaptive attacks. In the first paragraph of Section 4 (Experiments), we stated that *”We also provide an analysis of the detection performance under two types of adaptive attacks in Appendix B.8. We find that the attacker needs to sacrifice two crucial elements (attack success rate or stealthiness) to evade our detection method”*
>
> Concretely, we have tested two types of adaptive attacks: 1) *Increasing trigger visibility*, and 2) *Decreasing attack success rate (ASR)*. The first type of attack falls into the category of adaptive attacks suggested by the reviewer. The results show that significantly increasing the trigger size or the blending transparency could only evade a small amount of detection. Actively lowering the ASR, however, is more effective, which of course, will sacrifice the attack performance.
>
> Following the reviewer’s suggestion, we have also included a regularization-based adaptive attack in Appendix B.8. It regularizes the model to pay *uniform attention* (defined by the *input gradient*) to the trigger patterns.  Our results indicate that such a technique could make patch-based attacks (e.g., BadNets) easier to be detected by our CD. It can indeed evade our detection if applied on full-image trigger patterns (e.g., Nashville attack), however, the ASR would decrease to only 12.9%, leading to failed attacks. More details can be found in the updated paper (Appendix B.8).
>
> We have already tested the Trojan attack mentioned by the reviewer in our main results (Table 1 and Figure 3), the detection AUC by our CD-L method is >96%. In fact, 9 out of the 12 tested attacks adopt a full-image size trigger, as summarized in Table 4 Appendix B.3.
>
> ---
> **Q4:** Bias Detection
>
> **A4:**
> We believe that our contribution to bias detection is by no means trivial.  To the best of our knowledge, bias is a rather abstract concept that so far can only be identified manually [3]. Most existing works in this field only mitigate the known biases in a dataset, such as gender bias, color bias, or simply inter-class imbalances [4,5,6]. We believe the current literature is in great need of an automated bias detection technique that ideally can be applied to any type of DNNs. In this work, we demonstrate the effectiveness of our CD method for bias detection and its potential to serve as an automated bias mining technique.
>
> The other novelty of the bias detection experiment is that it closes the gap between two fields, i.e., bias detection and backdoor example detection. We believe our findings in this paper may also benefit dataset debiasing and fair machine learning. We note that bias detection is of course not the main focus of our paper and so our contribution is to highlight the potential of our method in this context.   The main contribution of our paper is the use of CD for backdoor sample detection.
>
> ---
>
> [3] Quanshi Zhang, Wenguan Wang, and Song-Chun Zhu. Examining cnn representations with respect to dataset bias. In AAAI, 2018.
>
> [4] Tolga Bolukbasi, Kai-Wei Chang, James Y Zou, Venkatesh Saligrama, and Adam T Kalai. Man is to computer programmer as woman is to homemaker? debiasing word embeddings. NeurIPS, 2016.
>
> [5] Junhyun Nam, Hyuntak Cha, Sungsoo Ahn, Jaeho Lee, and Jinwoo Shin. Learning from failure: De-biasing classifier from biased classifier. NeurIPS, 2020.
>
> [6] Enzo Tartaglione, Carlo Alberto Barbano, and Marco Grangetto. End: Entangling and disentangling deep representations for bias correction. In CVPR, 2021.

---

> ### Author Response · Authors · 2022-11-17
> **Any additional questions?**
>
> Dear reviewer czZL. Thanks again for reviewing our paper. Please let us know if you have any additional questions or require further clarification. We are happy to address them before the rebuttal ends.

---

> ### Author Response · Authors · 2022-11-23
> **Thanks for your initial comments**
>
> Dear reviewer czZL,
>
> Please allow us to express our gratitude for your initial comments. They have greatly helped improve our work. We hope our responses have clarified your concerns. Please kindly let us know if anything is unclear. Although we couldn’t update the paper at this stage, we will try our best to answer any remaining questions.

---

> ### Author Response · Authors · 2022-11-29
> **Your further feedback is much appreciated.**
>
> Dear Reviewer czZL,
>
> Thanks again for reviewing our paper. We have added new results and explanations to address the concerns about **1)** the novelty of our work (the difference to NC and [1]); **2)** the robustness to adaptive attacks (3 adaptive attacks including the suggested regularization-based adaptive attack); **3)** the potential and importance of extending our technique to bias detection; **4)** the assumption that backdoored model only uses a subset of the trigger (it is an empirical observation, i.e., the trigger is essential in its entirety across the dataset but only needs a small part to work on a particular input); and **5)** the existence of small but natural (rather than a backdoor, as detected by our method) patterns (such patterns exist but should be carefully reviewed for potential backdoor or bias).
>
> We are highly confident that our finding is generic for existing attacks (the 12 tested attacks at least), and it is not easy to force (or balance) the trigger to activate a large cognitive pattern on an unknown (uncontrollable) test image while being stealthy and effective at the same time. Intuitively, for an attack to evade our detection, the backdoor learning task should overwhelm the original learning task. In other words, the backdoor features should be complex enough to overtake the natural features during the learning process (in this case, the backdoor features will become the main features) while being invisible or unnoticeable to human observers. This arguably goes against the design principle of backdoor attacks: using a **simple** and **unnoticeable** trigger pattern to control the model's behaviors.
>
> Please kindly let us know if more clarifications or specific experiments are needed (we will include them in the next update). Your further feedback is much appreciated!

---

> ### Author Response · Authors · 2022-12-05
> **A kind reminder**
>
> Dear Reviewer czZL,
> We appreciate your initial comments on our submission. We have updated the paper and provided detailed clarifications for your concerns. But we haven't heard back from you. We understand that you might be very busy at the time of the year. But your further feedback would mean a lot to us. Please have a look at our response and kindly let us know if it has addressed your concerns. Thank you very much!

---

### Official Review · Reviewer_QeF1 · 2022-10-26

**Confidence:** 4
**Correctness:** 3
**Technical Novelty And Significance:** 2
**Empirical Novelty And Significance:** 3
**Recommendation:** 6

**Clarity, Quality, Novelty And Reproducibility:**

Clear exposition: well-organized demos and clear writing.

Easy to reproduce.

**Strength And Weaknesses:**

*************************
Strengths
*************************
+ Clear exposition: well-organized demos and clear writing.
+ Reasonable motivation.
+ Straightforward method.
+ Extensive experiments covering 3 datasets, 6 victim model architectures, 12 backdoor attacks, and 4 baseline detection methods. Results show their efficacy in terms of backdoor detection accuracy and face bias detection accuracy.
+ Easy to reproduce.

*************************
Weaknesses
*************************
- Technical novelty is incremental.
  - It borrows the idea form Neural Cleanse that embeds backdoor patterns in a minimal image subregion.

- The threat model is unreasonable.
  - The detection can only apply to local patterns. Simple global patterns, e.g., patterns in the frequency domain, can easily circumvent the detection.

- Experiments are unconvincing.
  - Backdoor pattern detection recall looks reasonable, but the precision is more measured. Please test on clean samples and report the false detection rate.

**Summary Of The Paper:**

The authors target to detect the input masks that extract the minimal backdoor patterns that lead to the model output. They further leverage the learned masks to detect and remove backdoor samples from poisoned datasets. Experiments show that they can detect several backdoor attacks and help detect biases from face datasets.

**Summary Of The Review:**

See the above.

---

> ### Author Response · Authors · 2022-11-10
> **Response to reviewer QeF1**
>
> Thanks for your valuable comments. We hope the following clarifications could address your concerns.
>
> ---
> **Q1:** Difference to Neural Cleanse (NC)
>
> **A1:** We clarify that the *only similarity* between our CD and Neural Cleanse (NC) is the L1 norm of the mask. Our CD differs from NC in multiple aspects:
> - NC is a method that detects **backdoor models** while our CD aims to detect **backdoor samples**. They are two different tasks.
> - NC finds a pair of pattern and mask that can perturb all samples from other classes into the target class, like a universal adversarial pattern [1] for the target class. Our CD finds a mask for an image that highlights the most essential pixels responsible for the model’s prediction. There is no “attack” in our CD approach.
> - NC takes a pre-trained model and a set of training images (and their labels), and outputs $K$ pairs of masks and trigger patterns, where K is the number of classes. However, backdoor sample detection requires a score for each sample, rather than $K$ pairs of masks and trigger patterns. NC can compute the score for each class (for backdoor label detection) but not for the samples.
> Our CD takes a pre-trained model and an image, and returns a corresponding mask without using any labels. This means that our CD can be applied at test time where there are no labels. Our CD is very efficient as its run time does not scale with the number of classes.
> - NC and CD have completely different optimization objectives (NC uses a cross-entropy loss defined w.r.t. a target label, while our CD uses the absolute difference w.r.t. original prediction). In terms of the mask, NC replaces the (1-m) area with clean samples from different classes, while our CD replaces the (1-m) area with $\delta$ that is randomly sampled at each optimization step.
> - NC does not work for attacks that use sample-specific triggers, which is not the case for our CD (as shown in Table 1 and Figure 3) which proves to be effective.
>
> We have also made a Figure to differentiate the two techniques, which can be found anonymously at [https://anonymous.4open.science/r/iclr-cd-403C/NC%20V.S.%20CD.png](https://anonymous.4open.science/r/iclr-cd-403C/NC%20V.S.%20CD.png).
>
> ---
>
> **Q2:** Frequency domain patterns.
>
> **A2:** Thanks for the suggestion. In fact, we have already evaluated one such **frequency domain** attack [2] in our initial submission. The attack is denoted as **Smooth** with the results reported in Tables 1/6/7/8/9 and Figures 2/3. Please note that we have used the trigger patterns released in their official repository for our experiments. Under lower poisoning rates, we find that the Smooth attack can evade all existing baselines (ABL/AC/SS/STRIP) but not our CD. We are happy to run more experiments if the reviewer has other suggestions.
>
> [1] Moosavi-Dezfooli, Seyed-Mohsen, Alhussein Fawzi, Omar Fawzi, and Pascal Frossard. Universal adversarial perturbations. In CVPR, 2017.
>
> [2] Yi Zeng, Won Park, Z Morley Mao, and Ruoxi Jia. Rethinking the backdoor attacks’ triggers: A frequency perspective. In ICCV, 2021.

---

> ### Author Response · Authors · 2022-11-10
> **Response to reviewer QeF1**
>
> ---
> **Q3:** Evaluation metric.
>
> **A3:** Thanks for the suggestion. We have added the area under the precision-recall curve (AU-PRC) to appendix C (Table 7 and Table 9 give detailed results on different models, attacks, and datasets) and summarized the result on CIFAR-10 (for training/test set, respectively) in the table below. It shows that our CD outperforms all baselines by an even larger margin, even more than was the case for the AU-ROC metric.
>
> The reason why we chose AU-ROC over AU-PRC is that the ultimate goal of backdoor sample detection is to remove all backdoor samples while minimizing the false positives. The area under the ROC curve aligns well with this objective. The results for AU-PRC can be found in the Appendix mentioned above.
>
> |  Attacks  |     ABL     |      AC     |  Frequency  |    STRIP    |      SS     |       CD-L      |     CD-F    |
> |:---------:|:-----------:|:-----------:|:-----------:|:-----------:|:-----------:|:---------------:|:-----------:|
> |  BadNets  |   26.13/-   | 38.99/22.50 | 36.67/32.47 | 60.81/57.70 | 38.94/18.01 | **75.53/75.48** | 60.09/60.18 |
> |   Blend   |   42.82/-   | 37.93/36.80 | 13.02/12.22 | 26.55/24.15 |  15.54/5.11 | **68.99/67.97** | 44.50/44.97 |
> |     CL    |   38.93/-   |  20.61/3.18 | 83.42/32.47 | 72.51/46.84 |  3.91/4.00  | **88.77/46.90** | 66.86/35.77 |
> |    DFST   | **49.19**/- | 48.12/46.86 | 24.57/23.21 |  5.40/5.48  |  25.85/6.48 | 47.34/**48.47** | 32.25/32.46 |
> |  Dynamic  |   46.65/-   | 48.15/45.53 | 75.03/74.76 | 64.35/59.29 | 21.92/10.72 | **88.89/87.65** | 70.95/69.09 |
> |     FC    |   39.88/-   | 51.80/79.71 | 82.50/87.78 | 38.57/39.61 | 38.09/14.78 | **89.48/90.34** | 72.74/81.15 |
> | Nashville |   26.24/-   | 63.88/39.04 |  11.76/9.54 |  5.70/4.59  | 30.57/10.57 | **89.44/87.60** | 66.46/65.02 |
> |    SIG    | **79.02**/- | 67.07/22.18 |  8.92/6.05  |  22.31/7.14 | 25.56/10.59 | 71.93/**55.14** | 71.11/62.72 |
> |   Smooth  |   27.41/-   | 53.83/40.09 |  5.90/5.10  |  10.49/8.72 | 41.67/12.75 | **55.66/52.26** | 24.34/23.32 |
> |   Trojan  |   26.30/-   | 25.07/24.78 | 63.81/62.89 | 59.38/59.35 |  15.58/4.96 | **80.56/80.12** | 60.54/61.85 |
> |   WaNet   |    5.69/-   | 37.77/38.48 | 61.22/61.01 | 28.01/25.16 | 30.78/14.82 | **63.58/64.99** | 44.51/47.47 |
> |    Avg    |   37.12/-   | 44.84/36.29 | 42.44/37.05 | 35.82/30.73 | 26.22/10.25 | **74.56/68.81** | 55.85/53.09 |

---

> ### Author Response · Authors · 2022-11-17
> **Any additional questions?**
>
> Dear reviewer QeF1,
>
> Thanks again for reviewing our paper. Please let us know if you have any additional questions or require further clarification. We are happy to address them before the rebuttal ends.

---

> ### Author Response · Authors · 2022-11-23
> **Further feedback**
>
> Dear reviewer QeF1,
>
> Thank you very much for your initial comments. Hope our clarifications and the new results can help address some of your concerns. Please kindly let us know if you have more questions. We will try our best to answer them before the rebuttal ends. We shall be grateful for any opportunity you would give us to improve our work.

---

### Author Response · Authors · 2022-11-10
**Updates in the revision**

We thank all reviewers for their constructive feedback. We have made the following major changes to address the reviewers' concerns:
- Added the area under the precision-recall curve (AU-PRC) in Appendix C, Table 7/9, where it shows that our CD outperforms the baselines by an even larger margin than the area under the ROC curve.
- Added the results for the Frequency detection [1] method to Table 1 and Figures 3/4.
- Updated the results in Figure 4 for mitigation adding additional finetuning steps. This improves the clean accuracy after mitigation.
- Added normalized false acceptance rate (N-FAR) in Figure 4, where a more noticeable improvement over the baselines can be observed.
- Added an additional adaptive attack in Appendix B.8 suggested by reviewer czZL

[1] Yi Zeng, Won Park, Z Morley Mao, and Ruoxi Jia. Rethinking the backdoor attacks’ triggers: A frequency perspective. In ICCV, 2021.

---

### Decision · Program_Chairs · 2023-01-20

**Decision:**

Accept: poster

**Justification For Why Not Higher Score:**

See the weakness mentioned above.

**Justification For Why Not Lower Score:**

The paper conducts solid experiments, and the method is well-designed if the assumption holds.

**Metareview: Summary, Strengths And Weaknesses:**

The paper proposes an algorithm to detect back-doored images by finding the "cognitive pattern" (CP), which is a minimum set of pixels responsible for the model's prediction. Based on the experiments on a wide range of backdoor attacks, the authors found that the images containing backdoor triggers often rely on small CP, and thus they proposed to use CP to detect images with backdoor triggers.

The review scores for the paper are pretty split, ranging from 3 to 8. After the zoom discussion, we identified the following strengths and weaknesses of the paper:

Strengths:
- The empirical results are pretty comprehensive in terms of evaluating many existing backdoor attacks, and the results suggest that the proposed algorithm is useful for defending against existing attacks.
- The proposed algorithm is well-motivated and easy to understand.

Weaknesses:
- The paper relies on the main assumption that adversarial triggers are small, and the model predictions on the back-doored images will rely on a small set of pixels. Although this is true for all the existing attacks, the assumption may not hold in general. For instance, it's possible to intentionally construct a large trigger and enforce the model to make predictions based on the whole trigger.
- Related to the previous point, we think an adaptive attack (by intentionally constructing such a trigger mentioned above) may break the proposed defense/detection. Although such an adaptive attack requires a very careful design.

Based on these weaknesses, we think the paper is borderline but finally decide to accept the paper due to its novelty and effectiveness (against standard backdoor attacks). We also compile a list of suggestions for the authors to improve their paper in the camera ready, as listed below:

- (Most Important!) The authors should state clearly their assumption that "the model predictions on the back-doored images will rely on a small set of pixels", so the readers may not expect the defense to be working against any kind of back-door attacks.
- To further improve the paper, the authors could conduct adaptive attacks to evaluate whether the proposed method can work with intentionally-designed large triggers. To ensure that the model focuses on the whole trigger instead of just a subset, the attacker can inject randomly masked triggers into the training set and associate them with the correct label. This will ensure that the model's prediction relies on the whole trigger instead of a subset of it.
- (minor) The proposed method is essentially trying to obtain a saliency map for the model's prediction. There are many other methods to obtain a saliency map in the interpretability community, and it's worthwhile to discuss or even compare the proposed method with other ways to generate the saliency map.




**Note From Pc:**

if the above contains the word "oral" or "spotlight" please see: "oral" presentation means -> notable-top-5% and "spotlight" means -> notable-top-25%. As stated in our emails, we are disassociating presentation type from AC recommendations

**Summary Of Ac-Reviewer Meeting:**

During the meeting, all the reviewers agree on the weaknesses mentioned above, although some reviewers think this is not crucial. We thus think this is a borderline case, with both pros and cons for the paper.